# Listen to Interpret: Post-hoc Interpretability for Audio Networks with NMF

**Jayneel Parekh**[1]    **Sanjeel Parekh**[1,2*]    **Pavlo Mozharovskyi**[1]
**Florence d'Alché-Buc**[1]    **Gaël Richard**[1]
[1]LTCI, Télécom Paris, Institut Polytechnique de Paris    [2]Audio Analytic
{jayneel.parekh,pavlo.mozharovskyi,florence.dalche,gael.richard}@telecom-paris.fr
sanjeel.parekh@audioanalytic.com

## Abstract

This paper tackles *post-hoc* interpretability for audio processing networks. Our goal is to interpret decisions of a trained network in terms of high-level audio objects that are also listenable for the end-user. To this end, we propose a novel interpreter design that incorporates non-negative matrix factorization (NMF). In particular, a regularized interpreter module is trained to take hidden layer representations of the targeted network as input and produce time activations of pre-learnt NMF components as intermediate outputs. Our methodology allows us to generate intuitive audio-based interpretations that explicitly enhance parts of the input signal most relevant for a network's decision. We demonstrate our method's applicability on popular benchmarks, including a real-world multi-label classification task.

## 1 Introduction

Deep learning models, while state-of-the-art for several tasks in domains such as computer vision, natural language processing and audio, are typically not interpretable. Their increasing use, especially in decision-critical domains, necessitates interpreting their decisions. A good interpretation is often characterized by its understandability for the end users (see for instance [16]). More importantly, attributes that aid understandability may largely be dependent on the data modality. In this paper, our aim is to generate *post-hoc* human–understandable interpretations for deep networks that process the audio modality. Here, *post-hoc* interpretability refers to the problem of interpreting decisions of a fixed pre–trained network.

Traditional approaches generate interpretations through input attribution, either directly on the raw input features or on a given simplified representation [30, 38, 36, 27]. To generate more understandable interpretations, a small number of approaches consider other means, such as logical rules [37], sentences [19] and high-level concepts [15].

Most existing *post-hoc* interpretability methods are primarily designed for application to images and tabular data. This limits their applicability to other data modalities such as audio. Although many audio processing networks operate on spectrogram-like representations, which can be seen as 2D time-frequency images, a visualization or attribution in this space is not as meaningful to a common user as it is for images [26].

This leads us to build an interpretation system that takes into account audio–specific understandability features. We motivate these features through an example: suppose an audio event detection network deployed in a house recognizes an "alarm" sound event. An ideal interpreter for this classification decision would have the ability to "show" that it was indeed an alarm sound that triggered this decision. To do so, it must be able to localize the alarm amid other events in the house (for *e.g.* dog

---

*Work conducted while the author was at Télécom Paris.

36th Conference on Neural Information Processing Systems (NeurIPS 2022).

barks, baby cries, background noise *etc.*) and make it listenable for the end–user. It is important to highlight here the role of listenable interpretations for better understanding of an audio network's decisions – note that it would be much less meaningful for a human to see the alarm sound as highlighted parts of a spectrogram. Thus making the following aspects important for our system design: (i) generating interpretations in terms of high–level audio objects that constitute a scene, (ii) segmenting parts of the input signal most relevant for a decision and providing it as listenable audio. It's worth emphasizing that audio interpretability is not the same as classical tasks of separation or denoising. These tasks involve recovering complete object of interest in the output audio. On the other hand, a classifier network might focus more on salient regions. When interpreting its decision and making it listenable we expect to uncover such regions and not necessarily the complete object of interest.

To this end, we propose a novel *post–hoc* interpreter for audio that employs a popular signal decomposition technique called Non–negative Matrix Factorization (NMF; Lee and Seung [25]). NMF seeks to decompose an audio signal into constituent spectral patterns and their temporal activations. Unlike principal component analysis, NMF is known to provide part–based decompositions [12]. Owing to these properties, we first use NMF to pre-learn a spectral pattern dictionary on our training data. This dictionary is then incorporated as a fixed decoder within our interpretation module. Specifically, we train our system to determine an intermediate encoding that performs two roles: (i) is able to reconstruct the input through the fixed NMF dictionary decoder, thus corresponding to time activations for dictionary components, (ii) at the same time, a function of this encoding is able to mimic the classifier's output. Training with these constraints allows us to generate, for any classifier decision, importance values over spectral patterns in our dictionary. Listenable interpretations are readily produced by inverting most important NMF spectral patterns back to the time domain.

In summary, we make the following contributions:

- We propose a novel NMF-based interpreter module for *post-hoc* interpretability that generates interpretations in terms of meaningful high–level audio objects, listenable for the end–user.

- We present an original formulation that constrains the interpreter encoding through two loss functions, one for input reconstruction through NMF dictionary and the other for fidelity to the network's decision. From a learning perspective, we show a new way to link NMF with deep neural networks, especially for generating interpretations.

- We extensively evaluate on two popular audio event analysis benchmarks, tackling both multi–class and multi–label classification tasks. The dataset for the latter is very challenging due to its collection in noisy real–world setting. Our method's design allows us to simulate feature removal and perform *faithfulness* evaluation.

## 2  Related Works

In this section, we position our work in relation to: (i) interpretability methods for audio, (ii) methods for concept–based interpretability and, (iii) use of NMF within the audio community, in particular, attempts to link it with deep networks.

**Interpretability methods for audio**  Some approaches [5, 44] have shown usability of attention/visualization techniques for interpreting audio processing networks or generated instance-wise feature importance [46] for time-series data [41]. However, our focus is on methods that attempt to address audio interpretability beyond image-based visualizations or raw input attribution. Muckenhirn et al. [31] illustrate usefulness of GuidedBackprop [40] for analyzing CNNs operating on raw 1D waveforms by analyzing relevance signal in frequency domain. This analysis does not extend to spectrogram input or address listenability of interpretations. A few works have applied the popular LIME algorithm with a simplified input representation more suited for audio. In particular, SLIME [28, 29] proposes to segment the input along time or frequency. The input is perturbed by switching "on/off" the individual segments. AudioLIME [18, 10] proposes to separate input using predefined sources to create the simplified representation. AudioLIME arguably generates more meaningful interpretations than SLIME as it relies on audio objects readily listenable for end-user. However, it can only be applied for limited applications as it requires existence of known and meaningful predefined sources that compose the input audio. APNet [47] takes another promising direction by

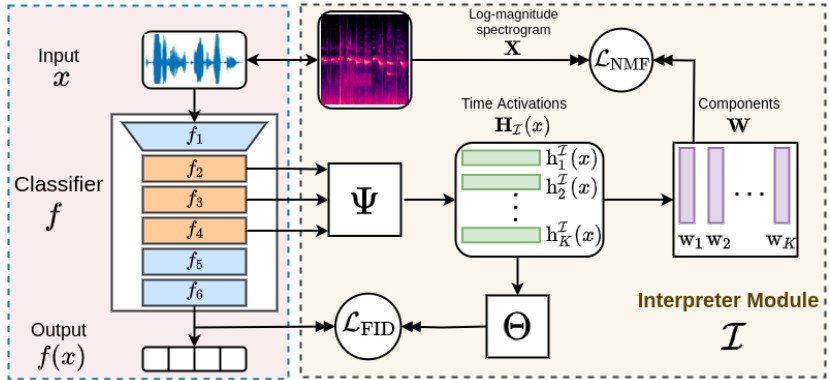

Figure 1: **System overview**: The interpretation module (right block) accesses hidden layer outputs of the network being interpreted (left block). These are used to predict an intermediate encoding. Through regularization terms, we encourage this encoding to both mimic the classifier's output and also serve as the time activations of a pre-learnt NMF dictionary.

extending interpretable prototypical networks for audio input. However, they propose an interpretable system *by-design*. They don't tackle the problem of *post-hoc* interpretation.

**Concept-based interpretability** Our method relies on high-level objects for interpretation. In this sense, it is most closely related to *post–hoc* concept-based methods [21, 15]. An interesting approach is that of *post-hoc* version of FLINT [32] with whom we share the idea of utilizing the hidden layers and loss functions to encourage interpretability. However we crucially differ from FLINT and other related approaches in concept representation and their applicability for audio interpretations. FLINT represents concepts by a dictionary of attribute functions over input space. The learnt concepts are not obviously comprehensible to a user, requiring a separate visualization pipeline to get insights. Approaches based on TCAV [21], such as ACE [15], ConceptSHAP [45], define concept using a set of images and learn a representation for it in terms of hidden layers of the network, termed as concept activation vector (CAV). These designs for concepts are not related to our NMF-inspired dictionary representation. Importantly, none of the above mentioned approaches can generate listenable interpretations which is key for understandability of audio processing networks.

**NMF for audio** has been widely used for numerous tasks ranging from separation to transcription [39, 42, 11, 6]. Its traditional usage as a supervised dictionary or feature learning method involves learning class-wise dictionaries over training data [12]. Time activations, which are the so-called features, are generated for any data point by projecting it onto the learnt dictionaries. Extracted features can subsequently be used for downstream tasks such as classification. Bisot et al. [7] couple NMF-based features with neural networks to boost performance of acoustic scene classification.

NMF has also been successfully employed with audio–visual deep learning models for separation [13] and classification [33]. Another line of research explored unfolding iterations of different NMF optimization algorithms as a deep neural network [23, 43]. These systems, commonly known as Deep NMF, have primarily been used for audio source separation tasks.

While these works share with us the idea of combining neural networks and NMF, there is no overlap between our goals and methodologies. Unlike aforementioned studies, we wish to investigate a classifier's decision in a post-hoc manner using NMF as a regularizer. Furthermore, to our best knowledge, attempting to regress temporal activations of a fixed NMF dictionary by accessing intermediate layers of an audio classification network is novel even within the NMF literature.

## 3 System Design

We begin this section with a brief note on data notation and NMF. Subsequently, in Sec. 3.1, we discuss interpreter module's design and learning. This is followed by a description of our interpretation generation methodology in Sec. 3.2. An overview of the proposed system is presented in Fig. 1.

**Data notation**. We denote a training dataset by $\mathcal{S} := (x, y)_{i=1}^{N}$, where $x$ is the time domain audio signal and $y$, a label vector. The label vector could be a one-hot or binary encoding depending

upon a multi-class or multi-label dataset, respectively. Since multiple audio representations are used in this paper, a note on their notation is in order. Very often, audio signals are processed in the frequency domain through a short-time Fourier transform (STFT) on $x$ called spectrogram. Log-mel spectrograms are a popular input to audio classification networks [34], which is also the one we use in this paper. To keep notation simple, we refer to input of the network with $x$. For NMF however, we favor a representation of $x$ that can be easily inverted back to the time-domain and use a log–magnitude spectrogram $\mathbf{X}$ that is computed by applying an element-wise transformation $x_0 \rightarrow \log(1 + x_0)$ on the magnitude spectrogram. This is preferred over using magnitude spectrograms as it corresponds more closely to human perception of sound intensity [17].

**NMF basics.** NMF is a popular technique for unsupervised decomposition of audio signals [3]. Given any positive time–frequency representation $\mathbf{X} \in \mathbb{R}_+^{F \times T}$ consisting of $F$ frequency bins and $T$ time frames, NMF decomposes it into a product of two non-negative matrices, such that,

$$\mathbf{X} \approx \mathbf{WH}$$

Here, $\mathbf{W} = [\mathbf{w}_1, \mathbf{w}_2, \ldots, \mathbf{w}_K] \in \mathbb{R}_+^{F \times K}$ is interpreted as the spectral pattern or dictionary matrix containing $K$ components and $\mathbf{H} = [\mathbf{h}_1, \mathbf{h}_2, \ldots, \mathbf{h}_K]^\intercal \in \mathbb{R}_+^{K \times T}$ a matrix containing the corresponding time activations. Typically, a $\beta$-divergence measure between $\mathbf{X}$ and $\mathbf{WH}$ is minimized and multiplicative updates are used for estimating $\mathbf{W}$ and $\mathbf{H}$ [25]. Note that it is possible to reconstruct signal corresponding to each or a group of spectral components. This is typically done using a procedure called soft–masking. For a single component $k$, this is written as,

$$\mathbf{X}_k = \frac{\mathbf{w}_k \mathbf{h}_k^\intercal}{\mathbf{WH}} \odot \mathbf{X}$$

Both ./. and $\odot$ are element-wise operations. If $\mathbf{X}$ is an invertible representation of the magnitude spectrogram, time domain signal for $\mathbf{X}_k$ is easily recovered using the inverse STFT operation. We extensively utilize this procedure for generating listenable interpretations. NMF can also be used for dictionary learning, by estimating $\mathbf{W}$ on a training dataset matrix $\mathbf{X}_{\text{train}}$. As discussed later, we use a variant of NMF called Sparse-NMF [24] to pre-learn dictionary for subsequent usage in the interpretation module.

### 3.1 Interpreter Design

As depicted in Fig. 1 the interpreter module $\mathcal{I}$ contains two components: an interpreter network and a NMF dictionary decoder. The so-called interpreter network computes the following function $x \mapsto \Theta \circ \Psi \circ f_{\mathcal{I}}(x)$ where $\Psi$ is the function responsible for generating an intermediate encoding from hidden layer representations of the classification network, and $\Theta$ attempts to mimic the classifier's output given the intermediate representation. The NMF decoder based on a pre-trained dictionary plays two roles: (i) during training, it constrains the intermediate representation to correspond to time activations of a pre-learnt spectral pattern dictionary and (ii) when interpreting a classifier's prediction, it is used to build listenable interpretation. To the best of our knowledge, this is an original usage of NMF that allows us to interpret a network's decisions in terms of a fixed dictionary.

**Design of $\Psi$.** The function $\Psi$ processes outputs of a set of hidden layers of the classifier, given by $f_{\mathcal{I}}(x)$. It's output, $\Psi(f_{\mathcal{I}}(x)) \in \mathbb{R}_+^{K \times T}$ produces an intermediate encoding of the interpreter. For simplicity, we will denote this intermediate encoding as $\mathbf{H}_{\mathcal{I}}(x) = \Psi \circ f_{\mathcal{I}}(x)$, a function over input $x$.

In practice, $\Psi$ is modelled as a neural network that takes as input convolutional feature maps from different layers of $f$. To concatenate and perform joint processing on them, each feature map is first appropriately transformed to ensure same width and height dimensions. Two important aspects were kept in mind while designing subsequent layers of $\Psi$. Firstly, audio feature maps for spectrogram-like inputs naturally contain the notion of time and frequency along the width and height dimensions. Secondly, through appropriate regularization we wish to produce an intermediate encoding that also serves as time activations of the pre-learnt NMF dictionary, a matrix of dimensions $K \times T$. To achieve this, we continuously downsample on the frequency axis and upsample the time axis to $T$ frames. Similarly, the number of input feature maps is re-sampled to reach a size of $K$, equal to the number of components in dictionary $\mathbf{W}$. Additionally, we ensure non-negativity of $\mathbf{H}_{\mathcal{I}}(x)$ through use of ReLU activation. All learnable parameters of $\Psi$ are denoted by $V_{\Psi}$.

**Design of $\Theta$.** $\mathbf{H}_{\mathcal{I}}(x)$, the intermediate encoding output by $\Psi$ is then fed to $\Theta$, which aims to mimic output of the classifier. This directly helps in learning a representation which can interpret $f(x)$. Its

design consists of two parts. The first part pools activations $\mathbf{H}_{\mathcal{I}}(x)$ across time. While this pooling can be implemented in multiple ways, we opt for attention–based pooling [20], *i.e.*, $\mathbf{z} = \mathbf{H}_{\mathcal{I}}(x)\mathbf{a}$, where $\mathbf{a} \in \mathbb{R}^T$ are the attention weights and $\mathbf{z} \in \mathbb{R}^K$ is the pooled vector. The pooled representation vector is passed through a linear layer. This is followed by an appropriate activation function to convert its output to probabilities, that is, softmax for multi-class classification and sigmoïd for multi-label classification. All learnable parameters of $\Theta$ are denoted by $V_\Theta$.

**Fidelity loss**. Generalized cross–entropy between interpreter's output $\Theta(\mathbf{H}_{\mathcal{I}}(x))$ and classifier's output $f(x)$ is minimized to encourage interpreter to mimic the classifier. For multi-class classification this loss function is written as,

$$\mathcal{L}_{\text{FID}}(x, V_\Psi, V_\Theta) = -f(x)^\intercal \log(\Theta(\mathbf{H}_{\mathcal{I}}(x))) \tag{1}$$

On the other hand, for multi-label classification this loss reads,

$$\mathcal{L}_{\text{FID}}(x, V_\Psi, V_\Theta) = -\sum f(x) \odot \log(\Theta(\mathbf{H}_{\mathcal{I}}(x))) + (1 - f(x)) \odot \log(1 - \Theta(\mathbf{H}_{\mathcal{I}}(x))). \tag{2}$$

Here $\odot$ denotes element-wise multiplication.

**NMF dictionary decoder and regularization**. We additionally constrain the intermediate encoding, such that, when fed to a decoder it is able to reconstruct the input audio. As already discussed, we choose this decoder to be a pre-learnt NMF dictionary, $\mathbf{W}$. Formally, through $\mathcal{L}_{\text{NMF}}$ we require $\mathbf{H}_{\mathcal{I}}(x)$ to approximate log-magnitude spectrogram $\mathbf{X}$ of input audio $x$ as $\mathbf{X} \approx \mathbf{W}\mathbf{H}_{\mathcal{I}}(x)$:

$$\mathcal{L}_{\text{NMF}}(x, V_\Psi) = \|\mathbf{X} - \mathbf{W}\mathbf{H}_{\mathcal{I}}(x)\|_2^2. \tag{3}$$

This allows us to consider $\mathbf{H}_{\mathcal{I}}(x)$ as a time activation matrix for $\mathbf{W}$.

**Training loss**. In addition to $\mathcal{L}_{\text{FID}}$ and $\mathcal{L}_{\text{NMF}}$, we impose $\ell_1$ regularization on $\mathbf{H}_{\mathcal{I}}(x)$ to encourage sparsity and well-behavedness, as is often done in classical NMF [24]. The complete training loss function over our training dataset $\mathcal{S}$ can thus be given as:

$$\mathcal{L}(V_\Psi, V_\Theta) = \sum_{x \in \mathcal{S}} \mathcal{L}_{\text{FID}}(x, V_\Psi, V_\Theta) + \alpha \mathcal{L}_{\text{NMF}}(x, V_\Psi) + \beta ||\mathbf{H}_{\mathcal{I}}(x)||_1 \tag{4}$$

where $\alpha, \beta \geq 0$ are loss hyperparameters. All the parameters of the system are constituted in the functions $\Psi$, $\Theta$ and dictionary $\mathbf{W}$. Since $\mathbf{W}$ is pre-learnt and fixed, the training loss $\mathcal{L}$ is optimized only w.r.t $V_\Psi, V_\Theta$. As a reminder, when training the interpreter for post-hoc analysis, the classifier network is kept fixed.

---

**Algorithm 1** Learning algorithm

---

1: **Input:** Classifier $f$, Training data $\mathcal{S}$, parameters $V = \{V_\Psi, V_\Theta\}$, hyperparameters $\{\alpha, \beta, \mu\}$, number of batches $B$, number of training epochs $N_{\text{epoch}}$
2: $\mathbf{W} \leftarrow$ PRE-LEARN NMF DICTIONARY $(\mathcal{S}, \mu)$                {// Sparse-NMF algorithm}
3: Random initialization of parameters $V_0$
4: $\hat{V} \leftarrow$ TRAIN $(f, \mathcal{S}, \mathbf{W}, V_0, \alpha, \beta, B, N_{\text{epoch}})$            {// Train with $\mathcal{L}$ in Eq. 4}
5: **Output**: $\hat{V} = \{\hat{V}_\Psi, \hat{V}_\Theta\}$

---

**Learning algorithm**. The complete learning pipeline is presented in Algorithm 1. The learnable parameters of the interpreter module are given by $V = \{V_\Psi, V_\Theta\}$. The pre-specified dictionary (Step 2 in Algorithm 1) is learnt using Sparse-NMF [24] wherein, the following optimization problem is solved through multiplicative updates to pre-learn $\mathbf{W}$:

$$\min D(\mathbf{X}_{\text{train}}|\mathbf{W}\mathbf{H}) + \mu\|\mathbf{H}\|_1 \quad s.t. \mathbf{W} \geq 0, \mathbf{H} \geq 0, \|\mathbf{w}_k\| = 1, \forall k. \tag{5}$$

Here $D(.|.)$ is a divergence cost function. In practice, euclidean distance is used. The reader is referred to appendix A.1 for more details regarding Sparse-NMF optimization problem, such as construction of $\mathbf{X}_{\text{train}}$ on our datasets.

## 3.2 Interpretation generation

Finally, to generate audio that interprets the classifier's decision for a sample $x$ and a predicted class $c$, we follow a two-step procedure: The first step consists of selecting the components which are

considered "important" for the prediction. This is determined by estimating their relevance using the pooled time activations in $\Theta$ and the weights for linear layer, and then thresholding it. Precisely, given a sample $x$, the pooled activations are computed as $\mathbf{z} = \mathbf{H}_{\mathcal{I}}(x)\mathbf{a}$. Denoting the weights for class $c$ in the linear layer as $\theta_c^w$, the relevance of component $k$ is estimated as $r_{k,c,x} = \frac{(\mathbf{z}_k \theta_{c,k}^w)}{\max_l |\mathbf{z}_l \theta_{c,l}^w|}$. This is essentially the normalized contribution of component $k$ in the output logit for class $c$. Given a threshold $\tau$, the selected set of components are computed as $L_{c,x} = \{k : r_{k,c,x} > \tau\}$.

The second step consists of estimating a time domain signal for each relevant component $k \in L_{c,x}$ and also for set $L_{c,x}$ as a whole. In this paper, we refer to the latter as the generated interpretation audio, $x_{\text{int}}$. For certain classes, it may also be meaningful to listen to each individual component, $x_k$. As discussed earlier under NMF basics, estimating time domain signals from spectral patterns and their activations typically involves a soft–masking and inverse STFT procedure. The inversion is performed using input audio phase $\mathbf{P}_x$. We detail this step with appropriate equations in Algorithm 2.

---

**Algorithm 2** Audio interpretation generation

---

1: **Input:** log-magnitude spectrogram $\mathbf{X}$, input phase $\mathbf{P}_x$ components $\mathbf{W} = \{\mathbf{w}_1, \ldots, \mathbf{w}_K\}$, time activations $\mathbf{H}_{\mathcal{I}}(x) = [\mathbf{h}_1^{\mathcal{I}}(x), \ldots, \mathbf{h}_K^{\mathcal{I}}(x)]^\intercal$, set of selected components $L_{c,x} = \{k_1, \ldots, k_B\}$.
2: **for all** $k \in L_{c,x}$ **do**
3: $\quad \mathbf{X}_k \leftarrow \frac{\mathbf{w}_k \mathbf{h}_k^{\mathcal{I}}(x)^\intercal}{\sum_{l=1}^K \mathbf{w}_l \mathbf{h}_l^{\mathcal{I}}(x)^\intercal} \odot \mathbf{X}$ $\qquad\qquad\qquad\qquad\qquad\qquad$ {// Soft masking}
4: $\quad x_k = \text{INV}(\mathbf{X}_k, \mathbf{P}_x)$ $\qquad\qquad\qquad\qquad\qquad\qquad\qquad\qquad$ {// Inverse STFT}
5: **end for**
6: $\mathbf{X}_{\text{int}} \leftarrow \sum_{k \in L_{c,x}} \mathbf{X}_k$
7: $x_{\text{int}} = \text{INV}(\mathbf{X}_{\text{int}}, \mathbf{P}_x)$
8: **Output:** $\{x_{k_1}, \ldots, x_{k_B}\}, x_{\text{int}}$

---

## 4 Experiments

We experiment with two popular audio event analysis benchmarks, namely ESC-50 [35] and SONYC-UST [9]. While the former is a multiclass environmental sound classification dataset, the latter appeared for DCASE 2019 and 2020 multi-label urban sound tagging task. We quantitatively and qualitatively evaluate different aspects of our interpretations, including a subjective evaluation carried out on SONYC-UST. The implementation of our system is available on GitHub[2]. This section is organized as follows: quantitative metrics and baselines are discussed in Sec. 4.1 followed by implementation details in Sec. 4.2. Experiments on ESC-50 and SONYC-UST are detailed in Sec. 4.3 and Sec. 4.4, respectively. We discuss some limitations in Sec. 4.5.

### 4.1 Quantitative metrics and baselines

**Metrics**. We quantitatively evaluate our interpretations in two ways. First, by evaluating how well it agrees with the classifier's output. For multi-class classification, this is done by computing fraction of samples where the class predicted by $f$ is among the top-$k$ classes predicted by the interpreter. We refer to this as the *top-$k$ fidelity*. To compute *fidelity* on multi-label classification tasks, we compute Area Under Precision-Recall Curve (AUPRC) based metrics between the classifier output $f(x)$ and its approximation by interpreter $\Theta(\mathbf{H}_{\mathcal{I}}(x))$. We compute macro-AUPRC, micro-AUPRC. Additionally, we report the maximum micro F1-score over different thresholds for the interpreter's output.

We also conduct a *faithfulness* evaluation for our interpretations. In general for any interpretability method, *faithfulness* tries to assess if the features identified to be of high relevance are *truly* important in classifier's prediction [1]. Since a "ground-truth" importance measure for features is rarely available, attribution based methods evaluate faithfulness by performing feature removal (generally by setting feature value to 0) and observing the change in classifier's output [1]. However, it is hard to conduct such evaluation for non-attribution or concept based interpretation methods on data modalities like image/audio, as simulating feature removal from input is not evident in these cases.

---

[2]https://github.com/jayneelparekh/L2I-code

Interestingly, our interpretation module design allows us to simulate removal of a set of components from the input. Given any sample $x$ with predicted class $c$, we remove the set of relevant components $L_{c,x} = \{k : r_{k,c,x} > \tau\}$ by creating a new time domain signal $x_2 = \text{INV}(\mathbf{X}_2, \mathbf{P}_x)$, where $\mathbf{X}_2 = \mathbf{X} - \sum_{l \in L_{c,x}} \mathbf{X}_l$. We define faithfulness of the interpretation to classifier $f$ for sample $x$ with:

$$\text{FF}_x = f(x)_c - f(x_2)_c \tag{6}$$

where $f(x)_c, f(x_2)_c$ denote the output probabilities for class $c$. It should be noted that this strategy to simulate removal may introduce artifacts in the input that can affect the classifier's output unpredictably. Also, interpretations on samples with poor fidelity can lead to negative $\text{FF}_x$. Both of these observations point to the potential instability and outlying values for this metric. Thus, we report the final faithfulness of the system as median of $\text{FF}_x$ over test set, denoted by $\text{FF}_{\text{median}}$. A positive $\text{FF}_{\text{median}}$ would signify that interpretations generally tend to be faithful to the classifier.

**Evaluated systems**. We denote our proposed Listen to Interpret (L2I) system, with attention based pooling in $\Theta$ by L2I + $\Theta_{\text{ATT}}$. The most suitable baselines to benchmark its fidelity are *post-hoc* methods that approximate the classifier over input space with a single surrogate model. We select two state-of-the-art systems, FLINT [32] and VIBI [4]. A variant of our own proposed method, L2I + $\Theta_{\text{MAX}}$, is also evaluated. Herein, attention is replaced with 1D max-pooling operation. Implementation details of the baselines are discussed in appendix A.7.

*Faithfulness benchmarking*: As already discussed, it is not possible to measure faithfulness for concept-based *post-hoc* interpretability approaches. While measurement for input attribution based approaches is possible, the interpretations themselves and the feature removal strategies are different, making comparisons with our system significantly less meaningful. We thus compare our faithfulness against a *Random Baseline*, wherein randomly chosen components are removed. To compare fairly, we remove the same number of components that are present in $L_{c,x}$ on average. This would validate that, if the interpreter selects *truly* important components for the classifier's decision, then randomly removing the less important ones should not cause a drop in the predicted class probability.

We also emphasize at this point that works related to audio interpretability (see Sec. 2), are not suitable for comparison on these metrics. Particularly, APNet [47] is not designed for *post-hoc* interpretations. AudioLIME [18] is not applicable on our tasks as it requires known predefined audio sources. Moreover, SLIME [29] and AudioLIME still rely on LIME [36] for interpretations. It is a feature-attribution method that approximates a classifier for *each* sample separately. As discussed before, these characteristics are not suitable for comparison on our metrics.

## 4.2 Implementation details

**Classification network**. We interpret a VGG-style convolutional neural network proposed by Kumar et al. [22]. This network was chosen due to its popularity and applicability for various audio scene and event classification tasks. It can process variable length audio and has been pretrained on AudioSet [14], a large-scale weakly labeled dataset for sound events. Further details about the network and its fine-tuning can be found in appendix A.2.

**Hyperparameters and training**. The hidden layers input to the interpreter module are selected from the convolutional block outputs. As is often the case with CNNs, the latter layers are expected to capture higher-order features. We thus select the last three convolutional block outputs as input to the network $\Psi$. For ESC-50, we pre-learn the dictionary $\mathbf{W}$ with $K = 100$ components and for SONYC-UST, we learn with $K = 80$ components. Reasons for choice of $K$ are discussed in appendix A.3. Ablation studies for other hyperparameters are in appendix A.4.

## 4.3 Experiments on ESC-50

**Dataset**. ESC-50 [35] is a popular benchmark for environmental sound classification task. It is a multi-class dataset that contains 2000 audio recordings of 50 different environmental sounds. The classes are broadly arranged in five categories namely, animals, natural soundscapes/water sounds, human/non-speech sounds, interior/domestic sounds, exterior/urban noises. Each clip is five-seconds long and has been extracted from publicly available recordings on the `freesound.org` project. The dataset is prearranged into 5 folds.

**Classifier performance**. The classifier achieves an accuracy of $82.5 \pm 1.9\%$ over the 5 folds, higher than the average human accuracy of 81.3% on ESC-50.

| System | Fidelity (in %) | | |
| --- | --- | --- | --- |
| | top-1 | top-3 | top-5 |
| L2I + $\Theta_{\text{ATT}}$ | $65.7 \pm 2.8$ | $81.8 \pm 2.2$ | $88.2 \pm 1.7$ |
| L2I + $\Theta_{\text{MAX}}$ | $73.3 \pm 2.3$ | $87.8 \pm 1.8$ | $92.7 \pm 1.2$ |
| FLINT [32] | $73.5 \pm 2.3$ | $89.1 \pm 0.4$ | $93.4 \pm 0.9$ |
| VIBI [4] | $27.7 \pm 2.3$ | $45.4 \pm 2.2$ | $53.0 \pm 1.8$ |

| System | Threshold $\tau$ | FF$_{\text{median}}$ |
| --- | --- | --- |
| | $\tau = 0.9$ | 0.002 |
| | $\tau = 0.7$ | 0.004 |
| L2I + $\Theta_{\text{ATT}}$ | $\tau = 0.5$ | 0.012 |
| | $\tau = 0.3$ | 0.040 |
| | $\tau = 0.1$ | 0.113 |
| Random Baseline | $\tau = 0.1$ | $< 10^{-4}$ |

Table 1: Quantitative results on ESC-50 environmental sound classification test data. (Left) top-$k$ fidelity (in %). FLINT and VIBI help benchmark fidelity but are not themselves suitable for audio interpretations. (Right) Faithfulness results (drop in probability) for different thresholds, $\tau$. We report FF$_{\text{median}}$ for proposed L2I + $\Theta_{\text{ATT}}$ and the Random Baseline.

**Quantitative results**. Mean and standard deviation of top-$k$ fidelity is calculated over the 5 folds. We show these results in Table 1 (Left) for $k = 1, 3, 5$. Among the four systems, VIBI performs the worst in terms of fidelity. This is very likely because it treats the classifier as a black-box, while the other three systems access its hidden representations. This strongly indicates that accessing hidden layers can be beneficial for fidelity of interpreters. FLINT achieves the highest fidelity, very closely followed by L2I + $\Theta_{\text{MAX}}$ and then L2I + $\Theta_{\text{ATT}}$. This experiment serves as a sanity check for our system, that while achieving fidelity performance comparable to state-of-the-art, we hold the advantage of providing listenable interpretations in terms of pre–learnt spectral patterns.

In Table 1 (Right), we report median faithfulness FF$_{\text{median}}$ averaged over the 5 folds for our primary system L2I + $\Theta_{\text{ATT}}$, at different thresholds $\tau$. Smaller $\tau$ corresponds to higher $|L_{c,x}|$, which denotes the number of components being used for generating interpretations. Thus, for Random Baseline, we report FF$_{\text{median}}$ at the lowest threshold $\tau = 0.1$, to ensure removal of maximal number of components. To recall the definition of Random Baseline, please refer to Sec. 4.1. FF$_{\text{median}}$ for L2I + $\Theta_{\text{ATT}}$ is positive for all thresholds. It is also significantly higher than the Random Baseline, indicating faithfulness of interpretations.

**Audio corruption experiment: an interpretability illustration**. We qualitatively illustrate that the interpretations are capable of emphasizing the object of interest and are insightful for an end-user to understand the classifier's prediction. To do so, we generate interpretations after corrupting the testing data for fold–1 in two different ways (i) either with white noise at 0dB SNR (signal-to-noise ratio), (ii) or mixing it with sample of different class. It should be noted that in both these cases the system is exactly the same as before and **not** trained with corrupted samples. Some examples, covering both types of corruptions are shared on our companion website.[3] A detailed qualitative analysis of this experiment can be found in appendix A.5 along with discussion about interpretations from other methods in appendix A.6.

## 4.4 Experiments on SONYC-UST

We now discuss experiments for the urban sound tagging task from the well known Detection and Classification of Acoustic Scenes and Events (DCASE) challenge 2019 & 2020 edition.

**Dataset**. The DCASE task used a very challenging real-world dataset called SONYC-UST [8]. It contains audio collected from multiple sensors placed in the New York City to monitor noise pollution. It consists of eight coarse-level and 20 fine-level labels. We opt for the coarse-level labeling task that involves multi-label classification into: 'engine', 'machinery-impact', 'non-machinery-impact', 'powered-saw', 'alert-signals', 'music', 'human-voice', 'dog'. This task is highly challenging for several reasons: (i) since it is real-world audio, the samples contain a very high level of background noise, (ii) the audio sources corresponding to the classes are often weak in intensity, as they are not necessarily close to the sensors, (iii) some classes may also be highly localized in time and more challenging to detect, (iv) lastly, noisy audio also makes it difficult to annotate, leading to labeling noise. This is especially true for training data that was labeled by volunteers.

**Classifier performance**. Our fine-tuned classifier achieves a macro-AUPRC (official metric for DCASE 2020 challenge) of $0.601$. This is higher than the DCASE baseline performance of $0.510$ and comparable to the best performing system macro-AUPRC of $0.649$ [2]. Note that it is obtained without use of data augmentation or additional strategies to improve performance.

---

[3]`https://jayneelparekh.github.io/listen2interpret/`

**Quantitative results**. In Table 2, we report the macro-AUPRC, micro-AUPRC and max-F1 for the interpreter output w.r.t classifier. For fairness, we ignore the class 'non-machinery impact' from all class-wise evaluations involved in fidelity (*i.e.* macro-AUPRC) or faithfulness. This is because the classifier predicts only one sample in test set with positive label for this class, causing AUPRC scores to vary widely for different interpreters. VIBI has the worst performance on all three metrics for this dataset as well. In contrast to ESC-50, here the best performing system is L2I + $\Theta_{\text{ATT}}$ followed by L2I + $\Theta_{\text{MAX}}$, and FLINT performing worse than both. The fidelity results on ESC-50 and SONYC-UST jointly demonstrate that our interpreter can generate high-fidelity *post-hoc* interpretations. Moreover, its design is flexible w.r.t different pooling functions.

The results for class-wise faithfulness are illustrated in Fig. 2a. We show $\text{FF}_{\text{median}}$ (absolute drop in probability) for our system and the Random Baseline. The results indicate that, for most classes, interpretations can be considered faithful, with a significantly positive median compared to random baseline results, which are very close to 0.

**Qualitative observations**. Qualitatively, we observe good interpretations for classes 'alert-signal', 'dog' and 'music'. For them, the background noise is significantly suppressed and the interpretations mainly focus on the object of interest. Interpretations for class 'human' are also able to suppress noise to a certain extent and focus on parts of human voices. However, for this class, we found presence of some signal from other audio sources too. For the remaining classes, namely 'Engine', 'Powered-saw' and 'Machinery-impact' the quality of the interpretation is more sample dependent. This is due to their acoustic similarity with the background noise. We provide example interpretations for SONYC-UST on our companion website.[3] We present an additional visualization to demonstrate coherence of our interpretations in appendix A.5.

**Subjective evaluation**. We perform a user study (15 participants) to evaluate quality and understandability of interpretations for L2I against SLIME on SONYC-UST test data. It is worth emphasizing that understandability is one important aspect of an interpretation but is not necessarily related to its faithfulness, which should be evaluated separately, for example as proposed in Sec. 4.1. As discussed earlier, SLIME is not suitable for comparison on our quantitative metrics. Nevertheless, it is the only relevant baseline for qualitative study of listenable interpretations. Details about SLIME implementation are in appendix A.7. A participant was provided with 10 input samples, a predicted class by the classifier for each sample and the corresponding interpretation audios from SLIME and L2I. They were asked to rate the interpretations on a scale of 0-100 for the following question: "*How well does the interpretation correspond to the part of input audio associated with the given class?*". The 10 samples were randomly selected from a set of 36 (5-6 random test examples per class). For each sample, we ensured that the predicted class was both, present in the ground-truth and audible in input. Class-wise preference results and average ratings are shown in Fig. 2b. L2I is preferred for 'music', 'dog' & 'alert-signal', SLIME is preferred for 'machinery-impact', no clear preference for others. A $t$-test with null hypothesis that the favourable system has a lower mean score yielded $p$-value $< 0.005$ for 'music', $< 0.05$ for 'dog' and 'machinery-impact' and $< 0.1$ for 'alert-signal'.

### 4.5 Limitations

Finally, we list below some limitations of this study: (a) Tuning the hyperparameters requires some experience with deep architectures and audio. (b) We use phase of original input spectrogram for time-domain inversion. One could employ a phase estimation algorithm to possibly improve over this strategy. (c) The current experiments are on two datasets and one network architecture. The design of the interpreter is dependent on task and architecture of base network. Our current design of $\Psi$ was proposed keeping in mind interpretations for a CNN operating on spectrogram-like representations. Nevertheless, it should be appropriately experimented with and modified when applying on any new data or network architecture.

### 5 Conclusion

To sum up, we have presented a system for post-hoc interpretation of networks that process audio. We posit that generating interpretations in terms of high-level audio objects and making them listenable are important attributes to aid understanding. Novel usage of NMF within our interpreter helps us satisfy both aforementioned requirements. Our original loss function formulation enables linking a classifier's decision to importance values over pre-learnt NMF spectral dictionary through an intermediate encoding. We perform extensive evaluation over popular audio event analysis datasets. We present a first-of-its-kind faithfulness evaluation for our non–attribution based method. Finally, a

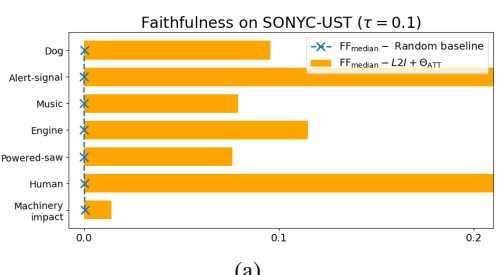
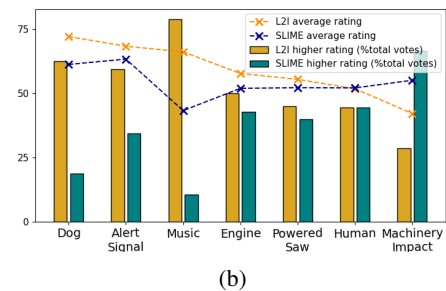

(a)                                                        (b)

Figure 2: (a) Faithfulness results (drop in probability) for SONYC-UST arranged class-wise for threshold, $\tau = 0.1$ (b) Subjective evaluation results. Average scores for L2I and SLIME and fraction of votes in favour of each system are reported.

|  | Fidelity | | |
|---|---|---|---|
| System | macro-AUPRC | micro-AUPRC | max-F1 |
| L2I + $\Theta_{\text{ATT}}$ | $0.909 \pm 0.011$ | $0.917 \pm 0.008$ | $0.847 \pm 0.010$ |
| L2I + $\Theta_{\text{MAX}}$ | $0.866 \pm 0.014$ | $0.913 \pm 0.012$ | $0.840 \pm 0.012$ |
| FLINT | $0.816 \pm 0.013$ | $0.907 \pm 0.011$ | $0.825 \pm 0.012$ |
| VIBI | $0.608 \pm 0.027$ | $0.575 \pm 0.019$ | $0.549 \pm 0.020$ |

Table 2: Fidelity results on SONYC-UST multi-label urban sound tagging task. We report AUPRC-based metrics and max F1 score for the interpreter w.r.t classifier's output (over three runs).

user study confirms usefulness of our listenable interpretations. Modular design of our system calls for further experimenting with decoder and other block architectures. We hope our work facilitates future research into designing modality-specific interpreters that aid understanding.

## Acknowledgments and Disclosure of Funding

This work has been funded by the research chair Data Science & Artificial Intelligence for Digitalized Industry and Services (DSAIDIS) of Télécom Paris. This work also benefited from the support of French National Research Agency (ANR) under reference ANR-20-CE23-0028 (LIMPID project). The authors would like to thank anonymous reviewers for their valuable comments and suggestions.

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
