# Listen to Interpret: Post-hoc Interpretability for Audio Networks with NMF

**Jayneel Parekh**[1]    **Sanjeel Parekh**[1,2*]    **Pavlo Mozharovskyi**[1]
**Florence d'Alché-Buc**[1]    **Gaël Richard**[1]
[1]LTCI, Télécom Paris, Institut Polytechnique de Paris    [2]Audio Analytic
{jayneel.parekh,pavlo.mozharovskyi,florence.dalche,gael.richard}@telecom-paris.fr
sanjeel.parekh@audioanalytic.com

## A    Appendix

### A.1    Sparse-NMF implementation details

The pre-specified dictionary (Step 2 in Algorithm 1) is learnt using Sparse-NMF [4]. To recall, the following optimization problem is solved through multiplicative updates to pre-learn $\mathbf{W}$:

$$\min D(\mathbf{X}_{\text{train}}|\mathbf{W}\mathbf{H}) + \mu\|\mathbf{H}\|_1 \quad s.t.\, \mathbf{W} \geq 0, \mathbf{H} \geq 0, \|\mathbf{w}_k\| = 1, \,\forall k. \tag{1}$$

Training audio files are converted into log-magnitude spectrogram space for factorization. We construct $\mathbf{X}_{\text{train}}$ differently for each dataset due to their specific properties. For ESC-50, $\mathbf{X}_{\text{train}}$ is constructed by concatenating the log–magnitude spectrograms corresponding to each sample in the training data of the cross-validation fold (1600 samples for each fold) and performing joint factorization using Eq. 1.

SONYC-UST however, is an imbalanced multilabel dataset with very strong presence of background noise. A typical procedure to learn components, as for ESC-50, yields many components capturing significant background noise. This affects understandability of interpretations. As a result, we process this dataset differently. We first learn $\mathbf{W}_{\text{noise}}$, that is, a set of 10 components to model noise using training samples with no positive label. Then, for each class, we randomly select 700 positively-labeled samples from all training data and learn 10 new components (per class) with $\mathbf{W}_{\text{noise}}$ held fixed for noise modeling. All $10 \times 8 = 80$ components are stacked column-wise to build our dictionary $\mathbf{W}$. While this strategy helps us reduce the number of noise-like components in the final dictionary, it does not completely avoid it.

As done in [1], for computational efficiency, we too average the spectrogram frames over chunks of five. This reduces the size of $\mathbf{X}_{\text{train}}$ and saves memory to allow training over more number of samples.

### A.2    Classifier $f$ details

The architecture we use for $f$ [3] has been pretrained on AudioSet. For each dataset, we first fine-tune this network and perform post-hoc interpretations for the resulting trained network. Here we discuss its broad architecture and specific training details used to fine-tune it on our datasets.

It takes as input a log-mel spectrogram. The architecture broadly consists of six convolutional blocks (B1–B6) and one convolutional layer with pooling for final prediction. Most convolutional blocks consist of two sets of conv2D + batch norm + ReLU layers followed by a max pooling layer.

Details of the full architecture can be found in the original reference. For fine-tuning, we modify the architecture of prediction layers. Specifically, we remove the F2 conv layer and add a linear

---

*Work conducted while the author was at Télécom Paris

36th Conference on Neural Information Processing Systems (NeurIPS 2022).

layer after final pooling, the output dimensions of which correspond to the number of classes in our datasets.

For both the datasets, we do not use any data augmentation. The ADAM optimizer [2] is used to fine-tune $f$. For ESC-50, we only fine-tune the prediction layers of the network. We train the classifier for 10 epochs on each fold of the dataset with a learning rate of $1 \times 10^{-3}$.

On SONYC-UST, we fine-tune all the layers in $f$, which leads to higher classifier AUPRC metrics. The classifier is trained for 10 epochs. Here we start with a learning rate of $2 \times 10^{-4}$ and halve it after every 4 epochs.

### A.3 Choosing number of components $K$

Choice of number of components, $K$, also known as order estimation, is typically data and application dependent. It controls the granularity of the discovered audio spectral patterns. Choosing $K$ has also been a long standing problem within the NMF community [9]. Our choice for this parameter was guided by three main factors:

- Choices made previously in literature for similar pre-learning of $\mathbf{W}$ [1], who demonstrated reasonable acoustic scene classification results with a dictionary size of $K = 128$. We used this as a reference to guide our choice for number of components.

- Dataset specific details which include number of classes, samples for each class, variability of recordings etc. For eg. acoustic variability of ESC-50 (larger number of classes), prompted us to use a dictionary of larger size compared to SONYC-UST.

- When tracking loss values for different $K$, we observed a plateauing effect for larger dictionary sizes as illustrated in Fig. 1 for ESC-50.

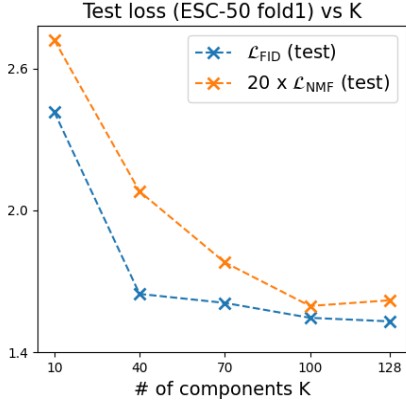

Figure 1: Loss values on ESC50-test data for fold 1 for various dictionary sizes.

### A.4 Other hyperparameters and ablation studies

**Audio processing parameters**. For both the tasks, we perform same audio pre-processing steps. All audio files are sampled at 44.1kHz. STFT is computed with a 1024-pt FFT and 512 sample hop size, which corresponds to about 23ms window size and 11.5ms hop. The log-mel spectrogram is extracted using 128 mel-bands.

**Other hyperparameters** We used the same set of hidden layers for both datasets. Specifically, we use the outputs of last three convolutional blocks in $f$, B4, B5 and B6. We also used the same loss hyperparameters $\alpha = 10, \beta = 0.8$ for both datasets. Models were optimized using ADAM [2] for 35 epochs on each fold of ESC-50 with learning rate: $2 \times 10^{-4}$ and for 21 epochs on SONYC-UST (learing rate: $5 \times 10^{-4}$).

Tab. 1 and Tab. 2 present ablation studies for loss hyperparameters and choice of hidden layers. The choices in bold indicate our current choices. The metrics and loss values given here are for a single

| ConvBlocks | $\mathcal{L}_{\text{NMF}}$ | $\mathcal{L}_{of}$ | top-1 |
|---|---|---|---|
| **B4+B5+B6** | **0.079** | **1.546** | **65.5** |
| B5+B6 | 0.103 | 1.572 | 61.5 |
| B6 | 0.118 | 1.698 | 57.8 |
| Input | 0.102 | 2.384 | 34.5 |

Table 1: Hidden layer ablation study (ESC-50). Current choice indicated in bold.

| $\alpha$ | $\beta$ | $\mathcal{L}_{\text{NMF}}$ | $\mathcal{L}_{of}$ | macro-AUPRC |
|---|---|---|---|---|
| **10.0** | **0.8** | **0.028** | **0.386** | **0.900** |
| 10.0 | 8.0 | 0.048 | 0.386 | 0.879 |
| 10.0 | 0.08 | 0.028 | 0.388 | 0.876 |
| 1.0 | 0.8 | 0.045 | 0.375 | 0.921 |
| 100.0 | 0.8 | 0.027 | 0.445 | 0.612 |

Table 2: Loss hyperparameter ablation study on SONYC-UST. Current choice in bold.

run. For the ablation study on hidden layers in Tab. 1, we additionally report another baseline where instead of accessing the hidden layers, $\Psi$ is directly applied on the input. Given that the interpreter no longer has access to representations learnt by the classifier (which were close to the output as well) and architecture of $\Psi$ itself is much simpler compared to the classifier, it is significantly worse at approximating classifiers output.

**Total training time** is around 50 minutes for 1 fold on ESC-50 and 150 minutes for SONYC-UST. Around 30-40% of the total time is spent on pre-learning $\mathbf{W}$ using Sparse-NMF (for both datasets). Networks were trained on a single NVIDIA-K80 GPU.

## A.5   Further discussion on Interpretations

### A.5.1   Corruption samples ESC-50

The goal of this experiment is to qualitatively illustrate that our method can generate interpretations on ESC-50 in various noisy situations. For this, we corrupt a given sample from a target class in two ways: (i) With sample from a different class (Overlap experiment), and (ii) Adding high amount of white noise, at 0dB SNR (Noise experiment). The key question that we want the interpretations to offer insight on is: *did the classifier truly make its decision because it "heard" the target class or is it making the decision based on the corruption part of the audio?* The cases where classifier misclassifies are analyzed in Sec. A.5.2. As already highlighted in Sec. 1, listenable interpretations are not expected to perform source separation for the class of interest, but to confirm if decision corresponds entirely/mostly to target class or not. All examples can be listened to on our companion website [2]. Since the target and corrupting signals and their classes are already known, we can reinforce the observations drawn by listening to the interpretations through spectrograms (Figs. 2, 3).

### A.5.2   Misclassification samples ESC-50

When the classifier prediction is incorrect, the interpretations may still provide insight into the classifier's decision by indicating what the classifier "heard" in the input signal. We give examples for this on the webpage[2]. For instance, one of the example is of a sample with ground-truth class 'Crying-Baby' misclassified as a 'Car-horn'. Interestingly, the interpretation is acoustically similar to car horns. Please note the importance of *listenable* interpretations that aid such understanding into the audio network's decisions.

### A.5.3   Coherence in interpretations

We qualitatively analyze the interpretations on SONYC-UST by visualizing relevances generated on the test set. Specifically, we compute the vector $r_{c,x} \in \mathbb{R}^K$ which contains relevances of all components in prediction for class $c$ for sample $x$. The relevance vectors are collected for each test sample $x$ and its predicted class $c$. We then apply a t-SNE [10] transformation to 2D for visualization. This is shown in Fig. 4. Each point is colored according to the class for which we generate the interpretation. Interpretations for any single class are coherent and similar to each other. This is to some extent a positive consequence of global weight matrix in $\Theta$. Moreover, globally it can be observed that classes like 'Machinery-impact' and 'Powered-Saw' have similar relevances which are to some extent close to 'Engine'. This is to be expected as these classes are acoustically similar. 'Dog' and 'Music' are also close in this space, likely due to the often periodic nature of barks or beats.

---

[2] https://jayneelparekh.github.io/listen2interpret/

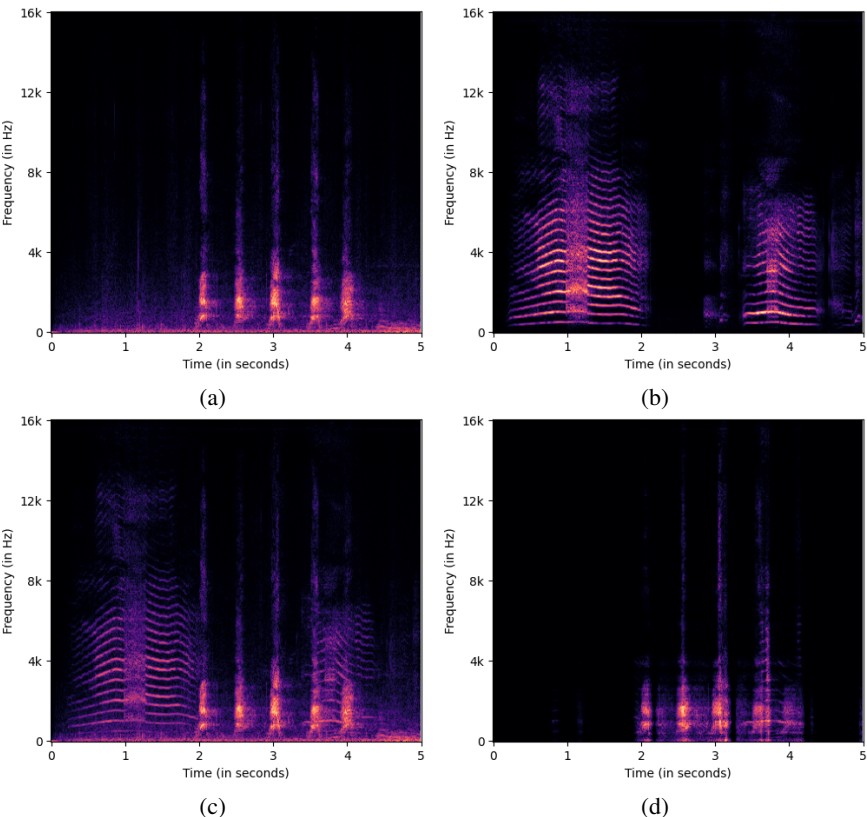

(a)

(b)

(c)

(d)

Figure 2: Log-magnitude spectrograms of an example from Overlap experiment: (a) Target class ('Dog') original uncorrupted signal (b) Corrupting/Mixing class ('Crying-Baby') signal (c) Corrupted/mixed signal, also the input audio to the classifier (d) Interpretation audio for the predicted class ('Dog'). The interesting observation is that spectrogram of interpretation audio almost entirely consists of parts from target class ('Dog') signal with only a very weak presence of corrupting class ('Crying-Baby') close to the end.

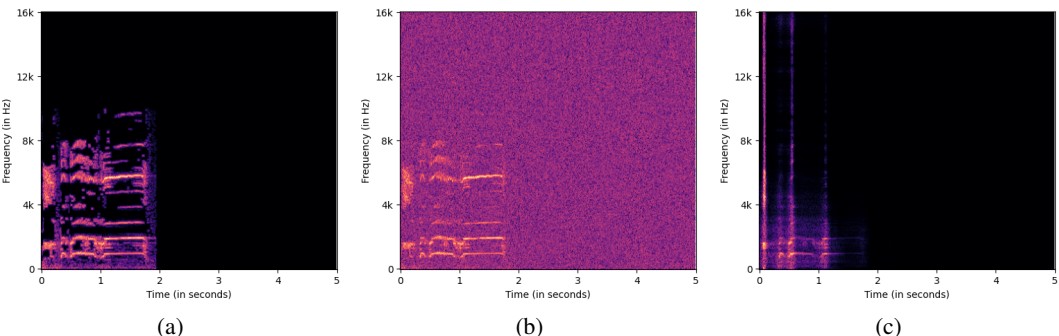

(a)

(b)

(c)

Figure 3: Log-magnitude spectrograms of an example from ESC-50 Noise experiment: (a) Target class ('Rooster') original uncorrupted signal, (b) White noise corrupted signal, also the input audio to the classifier (c) Interpretation audio for the predicted class ('Rooster'). Again, the interpretation audio is almost entirely free of corrupting signal (white noise in this case) and mostly consists of parts of the original target signal. This strongly indicates that the classifier relied on parts of audio corresponding to the target class to make its decision, and not the white noise.

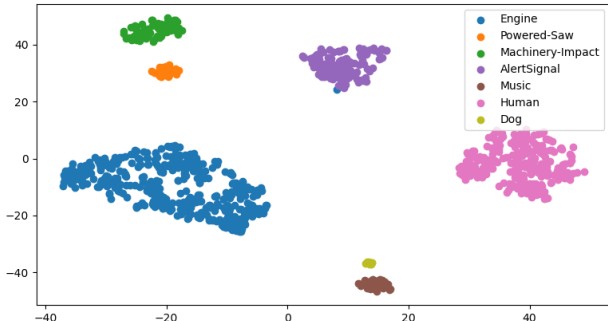

Figure 4: Visualized relevances (following a t-SNE transformation) of generated interpretations on SONYC-UST, colour-coded according to interpreted class.

## A.6 Discussion on interpretations from related methods

### A.6.1 Attribution maps for listenable output

Input attribution/saliency maps in their current form are more suitable for images. These maps are generally spatially smooth, which aids visual understandability, but are not effective masks to clearly emphasize time-frequency bins. Thus, for audio spectrogram like inputs, while they can be useful in visually indicating the important regions, they are poor masks to filter such information for listenable output. We applied a recent approach based on information bottleneck [8] to generate attribution maps for few samples on ESC50-Noise Experiment.

**Experimental details**: We used the python PyTorch version of their package and follow the standard example version given in their repository [3]. The example inserts a bottleneck in conv layer from 4th block of VGG16. Our network architecture is also similar to VGG architectures. So we applied a bottleneck at the output of 4th conv block (B4), which we also access via our interpreter. We also follow the same optimization procedure as in the example, i.e. Adam for 10 iterations. The saliency map is applied as a filter on the mel-spectrogram. We then approximate STFT from mel-spectrogram and invert it using input phase for a time-domain audio output.

Outputs can be heard on our companion website [2]. We provide visualizations for a sample in Fig. 5. While the saliency map indeed visually indicates relevant regions, the time-domain signal still contains considerable noise and is not very useful. The smoothness of saliency maps can be partly attributed to upsampling of information extracted from lower resolution feature maps. Another limitation of applying these methods to 2D CNN's is the frequent use of log-mel spectrogram as input (current model uses 128 mel bands) for the networks. The saliency map is then over the mel-spectrogram space. This adds to the loss of information and exacerbates issues in their use as filtering masks for spectrograms. Despite their usefulness, we believe these methods require non-trivial updates to be suitable for generating listenable interpretations.

### A.6.2 Interpretations of FLINT

For completeness, we also provide examples of interpretations by FLINT on ESC-50 Noise samples. As discussed in Sec. 2, FLINT uses a visualization pipeline to understand high-level attributes, which primarily consists of using activation maximization [5] based procedure to emphasize patterns relevant for the activation of an attribute.

In our current setting, this optimization procedure takes place in the log-mel spectrogram space. For initialization with a "weak version" version of the input we subtract 10 from the input log-mel spectrogram. We use Adam optimizer for 1500 iterations We add below examples of this visualization strategy after estimating log-magnitude spectrogram from the output of optimization procedure. Additionally we also estimate the time-domain signal as before to verify any potential as listenable output on the webpage [2].

---

[3]https://github.com/BioroboticsLab/IBA

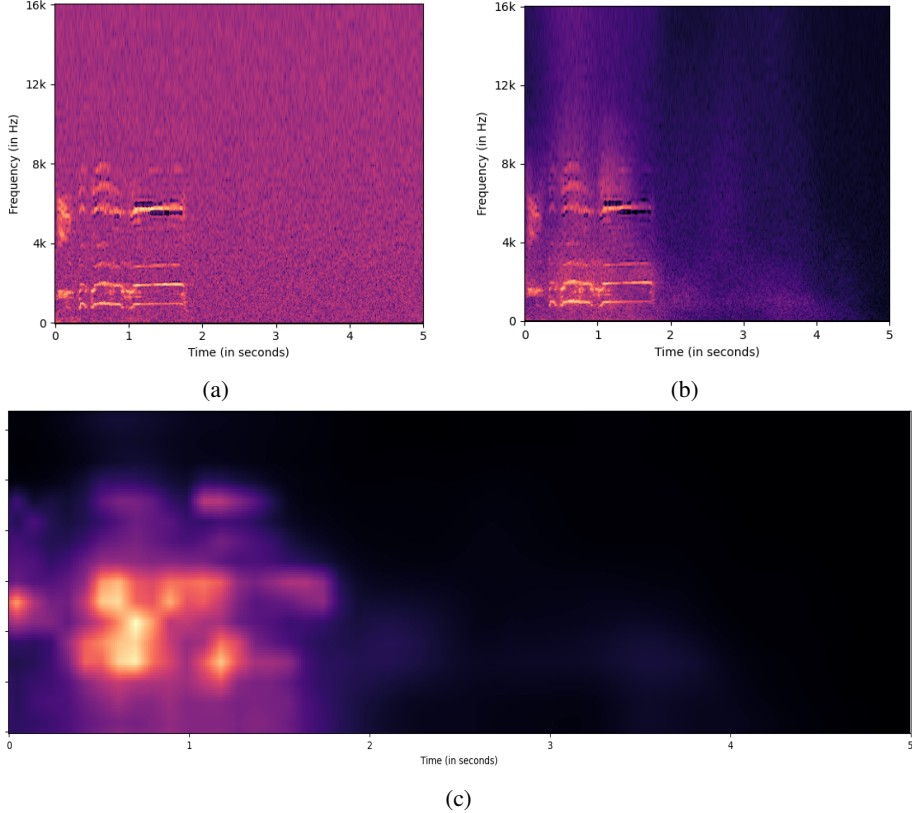

(a)                (b)

(c)

Figure 5: Log-magnitude spectrograms and saliency map to visualize an attribution map on ESC50-Noise sample: (a) White noise corrupted signal (from class 'Rooster'), also the input audio to the classifier, (b) Interpretation audio for the predicted class ('Rooster'), (c) Saliency map on the log-mel spectra space. The regions corresponding to the signal frequencies are brightest in the saliency map. However, owing to it's smoothness and loss of information in mel-spectrogram space, high amount of noise is still a part of interpretation signal.

The optimization in general results in specific patterns added in a log mel-spectrogram and thus the magnitude spectrogram. However, visually understanding the significance of the patterns is a very hard task. Listening to the resulting spectrograms is not informative either as they typically do not remove the noise, nor do they correspond to recognizable phenomenon. Compared to dictionary of pre-learnt spectral patterns, the dictionary of attributes is less constrained in the information an individual attribute encodes. Moreover, FLINT's visualization pipeline provides finer-grained interpretation at an attribute level. Both these considerations require the pipeine to be lot more effective to convey the interpretation understandably for audio modality.

### A.7    Baseline implementations details

**FLINT**: We implemented it with the help of their official implementation available on GitHub.[4] For each experiment, we fix their number of attributes $J$ equal to the number of our NMF components $K$. We also choose the same hidden layers for their system as we choose for ours. This baseline is trained for the same number of epochs as us. We use same values for our $\mathcal{L}_{\text{NMF}}$ loss weight, $\alpha$, and their $\mathcal{L}_{if}$ loss weight $\gamma$. For the other loss hyperparameters, we use their default values and training strategy.

**VIBI**: We implemented this using their official repository.[5] The key hyperparameters that we set are the input chunk size and their parameter $K$, the number of chunks to use for interpretation. We use a larger chunk size than in their experiments to limit the number of chunks. On ESC-50, we use a chunk size of $32 \times 43$, and on SONYC-UST, a chunk size of $32 \times 86$. This yields 40 chunks for each

---

[4]`https://github.com/jayneelparekh/FLINT`
[5]`https://github.com/SeojinBang/VIBI`

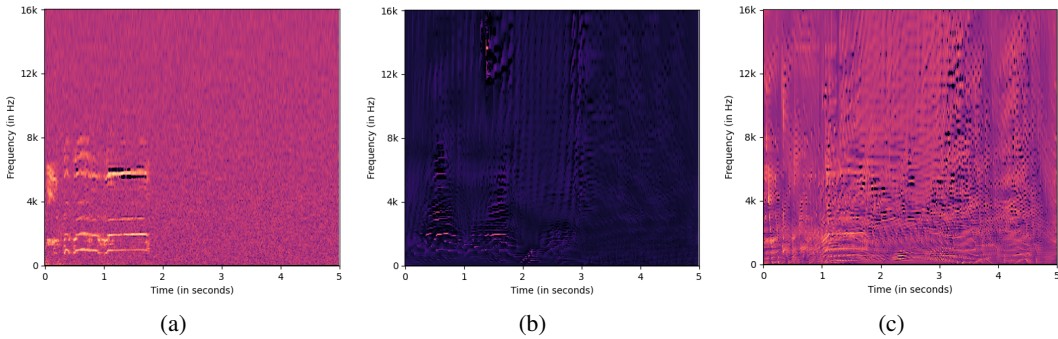

(a)                    (b)                    (c)

Figure 6: Log-magnitude spectrogram visualizations for two relevant attributes of FLINT on a sample from ESC50-Noise experiment: (a) White noise corrupted input audio (class: 'Rooster'), (b) Activation maximization output for attribute 62, (c) Activation maximization output for attribute 77.

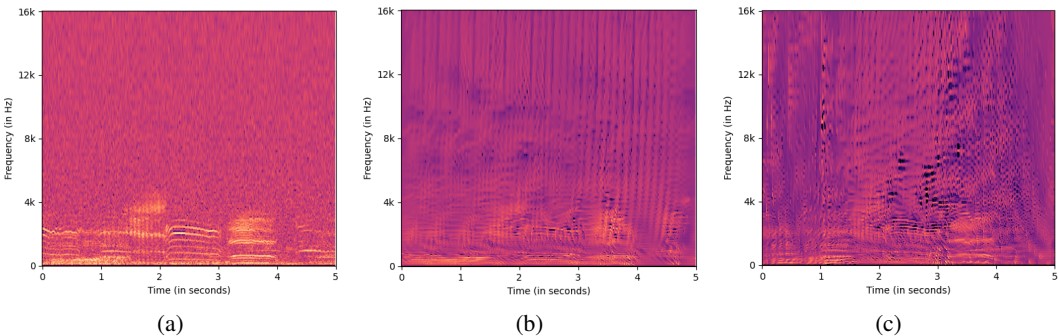

(a)                    (b)                    (c)

Figure 7: Log-magnitude spectrogram visualizations for two relevant attributes of FLINT on a sample from ESC50-Noise experiment: (a) White noise corrupted input audio (class: 'Sheep'), (b) Activation maximization output for attribute 7, (c) Activation maximization output for attribute 77.

input on both the datasets. We varied the $K$ from 5 to 20, and report the results with best fidelity. The system was trained for 100 epochs on ESC-50 and 30 epochs on SONYC-UST

**SLIME**: We primarily relied on implementation from their robustness analysis repository [6]. The key hyperparameters to balance are the number of chunks vs chunk size. SONYC-UST contains 10 second audio files. This is much longer than 1.6 second audio files for which SLIME was originally demonstrated [6]. Therefore, we divide only on the time-axis to limit the number of chunks. SLIME recommends a chunk size of at least 100ms. They operate on upto 290ms chunk size. We balance these two hyperparameters by dividing our audio files in 20 chunks of 500ms chunk size. We select a maximum of 5 chunks for interpretations and a neighbourhood size of 1000.

### A.8 Subjective evaluation implementation

The subjective evaluation interface was implemented using webMUSHRA [7]. Prior to voting on the test samples, participants were provided with an instruction page and then a training page with an example to get used to interface, instructions, tune their volume etc. Screenshots of the instruction and training page are given in Fig. 8, Fig. 9 respectively.

### A.9 Potential Societal Impacts

We expect our method to have positive societal impact by improving understandability of interpretations for audio processing networks. However, this inherently benign technology could be misused when in wrong hands. For example, it can be used to provide misleading interpretations if

---

[6]https://github.com/saum25/local_exp_robustness

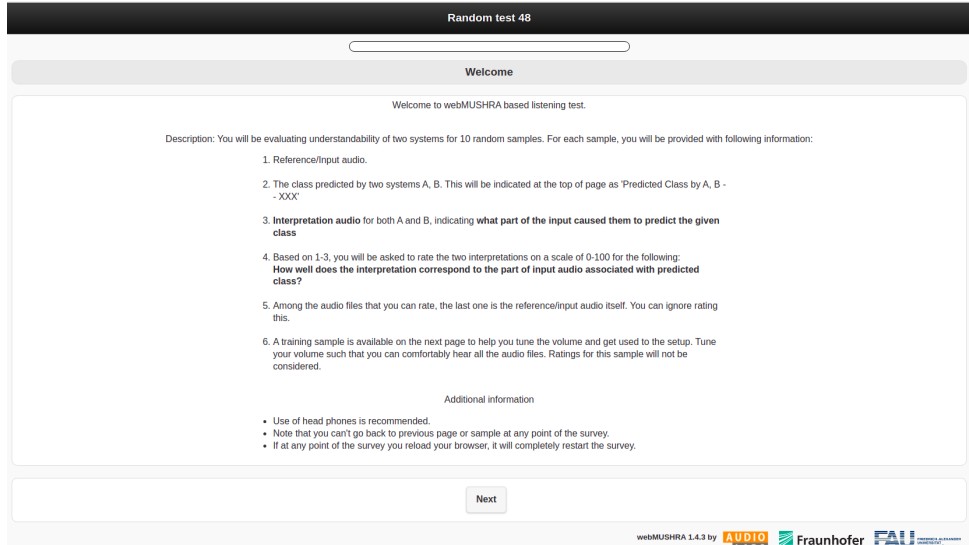

Figure 8: Instructions for the participants at the start of the subjective evaluation

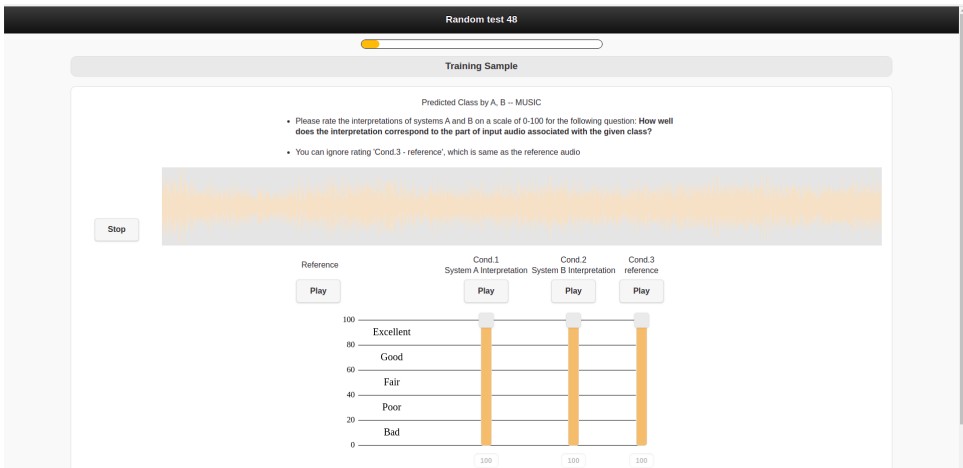

Figure 9: Training page for subjective evaluation that illustrates the interface for scoring for the participants.

trained incorrectly (wrong NN architectures, insufficient training examples/training epochs, malicious datasets etc.). Evidently, we expect proper use of the developed methodology, although direct misuse protection mechanisms were not developed in this piece of research, not being the initial goal.