# OpenReview forum: "Listen to Interpret: Post-hoc Interpretability for Audio Networks with NMF"
_NeurIPS.cc/2022/Conference — NeurIPS 2022 Accept_

### Official Review · Reviewer_GYnA · 2022-06-30

**Rating:** 6
**Confidence:** 4
**Soundness:** 3 good
**Presentation:** 3 good
**Contribution:** 3 good

**Summary:**

This paper aims to provide a methodology to interpret the decisions of neural networks that does classification in the audio domain. The designed methodology, 'listen-to-interpret' (L2I) is called to be 'post-hoc', which designates the fact that the proposed methodology is suitable to interpret the outputs of neural networks that have already been trained. The proposed methodology makes use of Non-Negative Matrix Factorization (NMF) to reconstruct the parts of the input that influence the decision the most. The intermediate representation layers of the original network is given to a learnable layer (psi) to estimate the temporal activation coefficients of a pre-trained NMF dictionary. The learnable layer that estimates the NMF temporal activation coefficients, as well a pooling layer+linear layer (theta) are trained to minimize the deviation between the NMF reconstruction and the input spectrogram, and to minimize the deviation between the classification output of the interpreter module and the original classification output given by the network.

The authors claim that the proposed methodology is preferable over the other concept based interpretability methods as the proposed method is able to generate audio. Also, compared to the other existing methods in the litterature, the authors claim that listen-to-interpret is a post-hoc interpretation method, as opposed to being an interpretable system by-design.

The authors quantitatively compare their method L2I against FLINT and VIBI in terms of the fidelity metric. The results seem to indicate that L2I outperforms VIBI, gives similar performance compared to FLINT. On the DCASE dataset, the authors also do a user study and show that 'listen-to-interpret' in general leads to better opinions.

**Questions:**

- More details should be given about how you used FLINT in your problem. I think it would be nice to show the interpretations obtained with FLINT. These are not audio signals, if I understand correctly, but it would be nice to see if the interpratations obtained with FLINT are useful or not. I think this is important as the faithfulness scores are similar with FLINT and listen to interpret. This can go in the appendix.

- If I am understanding correctly, you are simply using a learnable vector of length T to pool the NMF activation vectors over time. I have two concerns with regards to this:
	a) You are fixing the length of this vector. This means that in test time your system is limited to recordings of length T. Am I right?
	b) Did you consider for instance using something simpler such as calculating the average over the time axis? I think it would help to comment on why you opted use this pooling method over something simpler such as averaging. In fact in 1, it seems like your max-pooling based method seems to work better in terms of fidelity, compared to the attention-vector based method.

- In the in-line equation on $z=\sum_t H_I(x) a$ on page 5, the argument does not have a $t$ index. It would be worth to add an index over t, so that the notation is mathematically correct.

- In Figure 2, I would suggest using the same ordering with respect to classes in figures 1, 2.



**Limitations:**

I think this work presents a nice step towards postdoc-interpretability of audio-classification networks. However, one thing I am not super clear about is the motivation for using NMF. Is it because of the limited reconstruction ability of the linear model that enables the reconstruction of the input sound that is focused more on the salient parts of the input? One could imagine simply modifying the interpretation part of FLINT so that it construct audio e.g. with transposed convolutional layers?


**Strengths And Weaknesses:**

- Strengths:
	The paper proposes an interesting methodology that would be able to reconstruct the salient parts of the audio that supposedly affects the decision. In terms of the faithfulness metric, the reconstructed audio by the interpreter seems to outperform the VIBI approach and works about the same as the FLINT method.
	To the best of my knowledge, the fact that L2I is able to reconstruct audio for posthoc-interpretability is a plus, as the earlier methods are not capable of that.

- Weaknesses:
	The method seems to be quite similar to FLINT. Basically the main difference (if I am not missing anything) is the fact that you use NMF decoding to reconstruct the audio signal. This is not a big deal, as the usage of NMF seems to provide some benefit for interpretability. I think it would be good to know if the FLINT interpratations (even though it's not the audio modality) are useful at all. You can potentially show the obtained features with FLINT.

---

> ### Author Response · Authors · 2022-08-02
> **Response to Reviewer GYnA**
>
> We thank the reviewer for the positive feedback and are happy to respond to their questions and concerns. We first address the questions regarding FLINT and our usage of NMF. Then we address the questions pertaining to attention as our pooling method and the suggested writing changes.
>
> * **Relation of L2I with FLINT**: As we mention in line 88--93, Sec. 2, we are related to FLINT in the sense that both methods utilize intermediate layers of the classifier to extract high-level representations and use various loss functions to encourage properties related to interpretability. However, the functions modeling the interpreter, the extracted representations and its implications to interpretations are extremely different.
>
> * **Motivation for using NMF and Interpretations in FLINT**: The motivation for modelling representations in our interpreter like in NMF is that -- the factorization structure in NMF provides a meaningful way to decompose an audio signal. Once we are able to identify important components for the classifier's predicted class, the decomposition structure easily allows us to select those components and corresponding salient parts of input. This structure is not present in FLINT. FLINT extracts a dictionary of attributes/concepts from the hidden layers where each concept is a function from input space to $\mathbb{R}^{+}$. Even if one could very accurately reconstruct the input time-domain signal or STFT from this dictionary of concepts through the decoder, there is no decomposition structure to select parts of input from the importance over attributes. The interpretation in FLINT would still involve visualizing a attribute/concept in the input space (mel-spectrogram) through Activation Maximization like procedure. Yes, we can add these visualizations in the appendix.
>
> * **Attention vector and Pooling strategy**: The system is not limited to recordings of length $T$. This is because the attention vector $a \in \mathbb{R}^T$ is not modelled as a learnable vector but is the output of a network as in [1]. It essentially considers component activations for different time frames as a bag. It requires to know apriori the number of components, K but is not limited by size of the bag, i.e. number of time frames $T$. With regards to pooling, max-pooling is prone to spurious activation of components, which can help in better fidelity but are not useful for interpretations. On the other extreme, average pooling is robust to spurious activations but it can end up suppressing useful components if they activate only for a small period of time. Thus, we opted for a method that lies somewhere in between the two.
>
> * **Suggested writing changes**: Thanks for pointing out mistake in the equation. We will directly use $z=H_I(x)a$ everywhere. The matrix-vector product implicitly performs the summation over the time-frames. We will change Fig. 1 and use the same ordering of classes as in Fig. 2.
>
> [1] Maximilian Ilse, Jakub Tomczak, and Max Welling. Attention-based deep multiple instance learning. ICML 2018.

---

### Official Review · Reviewer_AaCp · 2022-07-03

**Rating:** 6
**Confidence:** 4
**Soundness:** 2 fair
**Presentation:** 3 good
**Contribution:** 3 good

**Summary:**

This paper proposes a system for interpreting deep audio classifiers. It works by creating audio excerpts of the parts of the input most relevant to the classifier's decision. To this end, a dictionary of spectral components is learned using NMF. Additionally, representations from a trained audio classifier are extracted and passed to an interpreter network that estimates activations for these NMF components. The interpreter is trained using a loss that encourages reconstruction of the input from the estimated activations and (fixed) spectral components as well as a loss for recovering the classifier's prediction from the activations. The system is evaluated on two audio classification datasets with regard to fidelity (how well does the interpreter recover the classifier's predictions?) and faithfulness (how well does the interpreter determine parts of the input that are relevant to the classifier's decision?). The proposed system admits a natural formulation for measuring faithfulness.

**Questions:**

I have a number of questions and concerns that should be clarified before accepting the paper.

Major:
- The proposed method contains learnable parameters. This raises the question: How much do the interpretations obtained depend on the dataset used for training the interpreter? Do the interpretations give insights into the classifier's behavior or are they merely artifacts of interpreter training? These questions should be openly discussed in the text. An interesting experiment to perform would be a cross-dataset experiment. How does the system perform when combining a ESC-50 classifier with a SONYC-UST-trained interpreter and vice-versa? Do we still obtain useful interpretations? How are fidelity and faithfulness impacted?
- In line 239, it is mentioned that for computing faithfulness in the multi-class case, changes in logit values instead of changes in probability are used. This makes no sense to me. This way, one can obtain a positive faithfulness value even thought the probability predicted for $c$ has increased for $x_2$ compared to $x$! Additionally, the use of logits here means that the $FF_{median}$ values reported in Table 1 (right) have no interpretable units. What does $1.29$ mean here? Is this a lot or is it little? Probabilities should be used for the multi-class case as well.
- The definition of the "Random Baseline" in line 256 is problematic. Here, components that are not in $L_{c,x}$ are removed which potentially makes this baseline overly pessimistic. It would make more sense (and correspond much more with the term "Random Baseline") if a random subset of all components was removed (not only those that are not in $L_{c,x}$). In addition, please report: What is the total number of components? What is the average size of $L_{c,x}$ for different $\tau$?
- The subjective listening test as described in this paper is highly questionable and should be removed. The test asked participants to rate the similarity between interpretations and sounds associated with the ground truth class. But that is not necessarily what the interpretations should sound like! The interpretations should correspond to whatever parts of the input habr been used for the classifier's prediction. These parts may or may not correspond to sounds associated with the ground truth class. The listening test presented here is unable to capture anything about the classifier's decision process. It is likely that the subjective scores correspond more to overall separation or sound quality. In addition, no kind of significance testing has been performed.

Minor:
- It is not clear from the main text of the paper how phase information is obtained to generate the listenable outputs from magnitude spectrograms. From Algorithm 2, which mentions "Input Phase", I presume that the phase of the original, unmodified input spectrogram is used. This should be clarified in the text and it should be emphasized that it will lead to artifacts.
- How do you ensure non-negativity of $H_\mathcal{I}(x)$ while minimizing $\mathcal{L}_{NMF}$? If $H_\mathcal{I}(x)$ is not constrained to be non-negative, the resulting decoder does not perform "true" NMF but rather semi-NMF. The resulting decomposition would not be truly part-based. This should be clarified in the text.
- It is unclear why the additional $l_1$ regularization on $H_\mathcal{I}(x)$ is used. What is the impact of this choice? Line 185 mentions "well-behavedness, especially for large dictionary sizes", but the reference given is for Sparse NMF, which solves a different optimization problem. In Sparse NMF, both $W$ and $H$ are unknown, whereas here, $W$ is fixed, right?
- Algorithm 1 is an overly mathy way of writing a very straightforward thing and should be dropped. It is clear from the text that the dictionary is learned prior to training the interpreter.
- As a baseline, it would be more interesting to compare the proposed architecture with a $\Psi$ that takes the input $x$ directly instead of intermediate representations. This way, the impact of utilizing intermediate representations would have been more clearly demonstrated. The paper does contain comparisons with some published baselines that directly work on the input $x$, but these baselines also differ from the proposed system in other aspects.
- Why is $\tau=0.1$ in Figure 2? What is the impact of $\tau$ here? For the other dataset, different values of $\tau$ were used.
- Section A.1.1 in the supplemental: It is mentioned that a special procedure was performed to reduce the number of noise-like components in the dictionary for SONYC-UST. What is the impact of this? Does it lead to better or worse fidelity or faithfulness? Do the generated interpretations sound more natural?
- Section A.1.3 in the supplemental: This is very confusing. Firstly, the number of components that is ultimately chosen for the proposed system should be mentioned in the main text and this section should then be referenced. Secondly, what is $\mathcal{L}_{of}$? Thirdly, are we looking at the test loss of the interpreter here? Why not simply look at the loss during dictionary learning and choose K depending on that?
- Line 58 in the supplemental: Which early stopping criterion has been used? In other words: how have the numbers 35 and 21 been obtained?
- Tables 1 and 2 in the supplemental: On which datasets have these values been computed?
- The accompanying website is quite nice, but rather than just showing the audio examples and spectrograms, it would be very important to also give the corresponding probabilities predicted by the original classifier. After all, we are not interested in seeing source separation performance here but in explaining the classifier predictions.
- Throughout the paper and supplemental material, the text in figures is too small (with the exception of most of the text in Figure 1). The text in Figure 2 of the main paper is much (!) too small.
- More care should be taken in preparing the bibliography. For example, different styles are used to refer to the DCASE workshop in [2] and [8], a URL is given for [25] but for no other bibliography item, references for some venues are formatted inconsistently (e.g. WASPAA [40], [41] or NeurIPS [1], [39]), ...

Additional points:
- $x$ is first defined in line 117 and then redefined in line 123. This is confusing.
- Line 146 refers to some $f_\mathcal{I}(x)$ which is only defined in line 154. This is confusing.
- Equations (1) and (2) should be dropped. Cross-entropy losses are textbook material.
- Contractions should be avoided in the text (e.g. "It is" instead of "It's" in line 40).

**Limitations:**

The discussion on limitations of the proposed method could be extended. It should be emphasized that the interpretations obtained by this method depend on the datasets used for training the interpreter components and networks $\Theta$ and $\Psi$. Additionally, this section should mention that the method has only been applied to one classifier architecture and two datasets and that it may behave differently in other scenarios. It is also worth noting that the proposed method is quite computationally expensive (taking 150 minutes to train for one of the datasets, see Section A.1.4).

With regard to societal impact, the biggest problems would probably arise not from misuse of the proposed method but from false confidence in the given interpretations. At this point, all interpretability methods for deep learning systems should be approached with care. This is especially true for systems that intrdoduce additional learnable parameters (as is the case for the present submission). This should be clarified in the text.

**Strengths And Weaknesses:**

This is a well written paper with a neat and original idea at its core. The proposed incorporation of NMF into an interpretability pipeline is interesting and the ability to obtain listenable interpretations is very useful. I particularly liked that the proposed system admits a very natural formulation for measuring faithfulness of interpretations. The proposed system, or some of the ideas behind it, are likely to be used by other works in the future.

I do have several questions about technical details and concerns about some of the results presented. The experimental results are not totally convincing in their current form and one evaluation (the listening test) is highly problematic and should be removed altogether.

Nevertheless, if the points below are addressed, I think this could be a valuable contribution. For now, I recommend a borderline accept.

---

> ### Author Response · Authors · 2022-08-02
> **Response to Reviewer AaCp (Part 1/2)**
>
> We thank the reviewer for their feedback and detailed comments. We organize our answer as two separate comments. In this first one, we address the major concerns and in the second, we address the minor concerns:
>
> * **Interpreter and classifier + cross-dataset experiment**: Yes, interpretations depend on the dataset used to train the interpreter and classifier. The use of classifier's hidden layers (close to the output) by the interpreter and fidelity loss term, creates an explicit dependence between the two, providing insights into classifier's behavior. The interpreter is incentivized to rely on same features in the hidden layers for its output, as used by the classifier. Moreover, faithfulness results indicate that parts of input identified to be relevant to predicted class truly affect the classifier's output.
>     Due to the above reasons, it is not feasible to attach an interpreter of one dataset to classifier of other in our current setting. More specifically, (i) the classifier is fine-tuned on any dataset before we employ our system. Thus the intermediate layers used by interpreters are not the same for different datasets (they are same dimension but not same functions); (ii) the outputs of the classifier and interpreter from different datasets are not compatible. Thus it is hard to measure fidelity or faithfulness.
>     Nevertheless, the idea of cross-dataset experiments dwells around very interesting research topics of domain adaptation or transfer learning for interpretations. They have been increasingly explored for other learning problems, but are still open-ended for interpretation methods involving learning. The suggested experiment is outside our current scope but could be taken up as a future work.
>
> * **Faithfulness evaluation + Random Baseline**: We used logit values to better emphasize the change in classifiers output without any influence of other classes. However, we can report the probability drop as well. The paper currently contains $FF_{median}$ over ESC-50 (fold 1) for $\tau$ ranging from $0.9$ to $0.3$. We will update these results by reporting average of $\text{FF}_\text{median}$ over 5 folds for absolute logit drop and absolute probability drop from threshold $0.9$ to $0.1$. We also updated the 'Random Baseline' to randomly select components to remove from among all the components. All these results results are given in a table below. Our system is in the first five rows. The last row is for 'Random Baseline':
> | Threshold $\tau$  | $FF_\text{median, logit}$  | $FF_\text{median, prob}$  | $size(L_{c,x})_{mean}$ |
> | ------------- |:-------:| -------:| -------:|
> | $\tau=0.9$    | 0.336 | 0.002 | 1.208 |
> | $\tau=0.7$    | 0.526 | 0.004 | 1.762 |
> | $\tau=0.5$    | 0.854 | 0.012 | 2.676 |
> | $\tau=0.3$    | 1.374 | 0.040 | 4.096 |
> | $\tau=0.1$    | 2.126 | 0.113 | 6.871 |
> | ------- |-----|----|----|
> | $\tau=0.1$    | 0.015 | $<10^{-4}$ | 7 |
>
> The average number of components used $|L_{c,x}|_{mean}$ is small compared to dictionary size on ESC-50 ($K=100$) which is why updated result for 'Random Baseline' still has only a small positive median.
>
> * **Subjective Evaluation**: Throughout the paper we only generate interpretations for classes predicted by the classifier. So indeed, we generate interpretations for the classifier's decision. This is true for faithfulness experiments, qualitative examples on the website and subjective evaluation. Yes, you are right that interpretations should correspond to whatever parts of the input have been used for the classifier's prediction. However the "ground-truth interpretations" are not available so we cannot measure if our/SLIME's interpretation is closer to what the classifier 'truly' based its decision on. We only intended to measure quality and understandability of the interpretations (line 350, Pg 9) which is also an important aspect of interpretability. We can add this as a disclaimer to the evaluation. We can add significance testing results. We report 4 classes with a clear favourable system. 'Dog', 'AlertSignal', 'Music' for L2I and 'MachineryImpact' for SLIME. For these classes, we performed a t-test with the null hypothesis that the favourable class has a lower mean score. We obtained a p-value of $<0.005$ for 'Music', $<0.1$ for 'AlertSignal' and $<0.05$ for 'Dog', 'MachineryImpact'.

---

> ### Author Response · Authors · 2022-08-02
> **Response to Reviewer AaCp (Part 2/2)**
>
> The first comment (Part 1/2) addresses the major concerns. We address the minor concerns in this second comment (Part 2/2):
>
> * **Phase information**: Yes, we use input spectrogram phase for our time-domain inversions. We will explicitly mention it in Sec 3.2 and add a small discussion about this in limitations. Please refer to last point to reviewer iYoX for more details.
>
> * **Non-negativity of $H_I(x)$**: $H_{I}(x)$ is output of neurons with ReLU activation which ensures its non-negativity. We will add this detail in the main text.
>
> * **$\ell_1$ regularization on $H_I(x)$**: Yes, $W$ is fixed in our system and the two optimization problems (ours and SparseNMF) are not same. However, reconstruction of input through $H_I(x)$ is still common to both. $H_I(x)$ is still a large matrix owing to large number of components and time frames, and in both cases we expect a similar behaviour of time activations from the point of view of decomposing the input signal. Small number of components activating for any time-frame is still favourable for our problem. And thus, following SparseNMF loss we added the $\ell_1$ regularization.
>
> * **Baseline with $\Psi$ on input**: We created a system with exactly the same loss function, where input to $\Psi$ is $x$ and tried to make as little changes as possible in its architecture compared to our original $\Psi$. It achieves poor performance (top-1 fidelity on ESC50-fold1: 34.5\%). We can add its results in appendix for illustrative purposes but it's not so meaningful as design of $\Psi$ processes many intermediate maps with low resolutions. $x$ has just 1 map with full time resolution.
>
> * **Impact of $\tau$ and chosen values for different datasets**: Class-wise results were hard to show for ESC50 (50 classes) so we showed it for multiple thresholds. For SONYC-UST, multiple thresholds would have taken significant space so we chose a reasonable $\tau$ value ($\tau=0.1$ in Fig. 2). $\tau \rightarrow 1$ extracts only one relevant component. Too small $\tau$ extracts components not really important for prediction and capturing too many parts of the audio. A reasonable value is somewhere around 0.1 to 0.3 (on both datasets).
>
> * **Procedure to reduce noise-like components on SONYC-UST**: The fidelity and faithfulness were slightly worse for system not employing this procedure. However the main reason for the procedure was that interpretations often contained components which were constantly capturing background noise and affecting the understandability of interpretations. The procedure did not completely eliminate such components as some classes have noise-like characteristics but helped mitigate the issue.
>
> * **Stopping criterion for training**: The training output fidelity loss plateaus after around 12 epochs for SONYC-UST and around 25-30 epochs for ESC-50. The reconstruction loss plateaus earlier. Thus, not observing any significant performance gain or drop, we stopped the training a few epochs after that.
>
> * **Datasets for Tables 1 and 2 in supplement**: Table 1 is for ESC-50 (fold-1), Table 2 is for SONYC-UST. We will add this in the captions.
>
> * **Predicted probabilities on website**: We generate interpretations only for the predicted classes by the classifier. We explicitly mention the predicted class for all ESC-50 examples. The examples for SONYC-UST are also organized according to when the class was predicted by the classifier. We can make it more explicit for SONYC-UST and also add the predicted probabilities (for both ESC-50, SONYC-UST) as additional information.

---

> ### Comment · Reviewer_AaCp · 2022-08-08
> **Response to Authors**
>
> Thank you for your responses! You have addressed most of my concerns. As a consequence, I have bumped up my review rating from 5 (Borderline Accept) to 6 (Weak Accept).
>
> I have some questions remaining regarding Section A.1.3 in the supplemental (see also my initial review). I have still not understood:
> - What is $L_{of}$? The main text of the paper only mentions $L_{FID}$ and $L_{NMF}$.
> - What does Figure 1 in the supplemental show? Is this the test loss of the interpreter?
> - If yes, why is the loss of the interpreter used to decide on $K$? Why not simply look at the loss of Sparse-NMF during dictionary learning (Section A.1.1) and choose $K$ depending on that?
>
> Aside from this, some additional comments regarding your responses:
> - Part 1/2 - Faithfulness evaluation + Random Baseline: Thank you! I believe these numbers and the 'new' random baseline are much easier to understand.
> - Part 1/2 - Subjective Evaluation: As you mention, this experiment is only able to measure quality and understandability of the interpretations, but it does not say anything about faithfulness. Faithful interpretations may in fact be less understandable. At the same time, high quality interpretations may encourage false confidence in a user, even though the interpreter system is not very faithful. Thus, I would still suggest to remove this subjective evaluation. However, I understand your reasoning and agree that a disclaimer should be added, if the experiment is kept in the paper. The significance testing results could be mentioned briefly.
> - $l_1$ regularization on $H_I(x)$: In this case, may I suggest to add a comment along the lines of: "We use $l_1$ regularization to encourage sparsity of $H_I(x)$, as is sometimes done in classical NMF (cf. Sparse-NMF)."
> - Baseline with on $\Psi$ input: Thank you for this result. I believe adding this to the supplemental would strengthen the claim that intermediate representations are essential. But I agree that it is not essential to show.
> - Procedure to reduce noise-like components on SONYC-UST: Thank you. These explanations where very helpful to understand the impact of that procedure. I strongly suggest this explanation should be added to the supplemental.
> - Predicted probabilities on website: Adding the predicted probabilities would allow the reader to compare the probabilities predicted on the inputs and the interpretations. That would be helpful additional information.
> - Regarding the responses to other reviews: I also agree that showing visualizations for the other baseline methods (e.g. FLINT) would be useful.

---

> > ### Author Response · Authors · 2022-08-08
> > **Response to remaining queries (Reviewer AaCp)**
> >
> > Thank you for your positive update. Please find our response to your queries below:
> > * $L_{of}$ is a typo from us, it should be $L_{FID}$. Thanks for pointing this out. '$L_{of}$' was our old notation for fidelity loss.
> > * Yes, it is the test loss of interpreter in Fig. 1. The training loss also follows a similar trajectory. Tracking the reconstruction loss during dictionary learning phase for different $K$ could also be a helping factor in determining $K$. However, we didn't do it for two reasons: (i) we won't know the fidelity loss for different $K$ and (ii) The dictionary learning and interpreter training optimization problems are formulated and solved differently, thus it was a safer choice to track directly losses during the interpreter's training.
> > * We will take into account your other suggestions for the final version.

---

### Official Review · Reviewer_iYoX · 2022-07-11

**Rating:** 7
**Confidence:** 4
**Soundness:** 3 good
**Presentation:** 4 excellent
**Contribution:** 3 good

**Summary:**

This paper introduces a novel method for post-hoc interpretability of a trained deep audio classifier in order to exhibit what components of a spectrogram is majorly used in the decision-making. To do so, the authors propose to rely on a pre-trained dictionary of source components learned through a NMF decomposition trained over the original dataset. Their model then tries to use the activation of the network as a conditioning information to learn activation matrices of the corresponding dictionary. In order to ensure that this "interpreter" converges to an adequate solution, the learning objectives are both to reconstruct the original spectrogram, and also to deduce a similar representation to that of the final layer of the pre-trained network.

**Questions:**

- I feel that the bottleneck of your proposal lies in both the properties of the components (W) matrix. The quality of the results seem to be bounded by the cardinality of this set ?
- Could you provide more details on the classifier performance (sec. 4.3) and also shortly discuss the applicability of your method to other types of networks ?
- It is hard to gauge the result based on the "FF" (since it appears that it is a home-crafted metric), so the evaluation appears weakened. Could you provide a range of the FF as it is hard to know if "1.29" is a good result (since your baseline gives 0.0)

**Limitations:**

- You do not discuss at all the impact of the phase in your inversion. From what I understood, you simply use the phase information of the original spectrogram ? However, you could still have some incoherence depending on what you select, and if your masking information is a combination of several external components of W. Wouldn't it make sense to perform at least a few iterations of a phase reconstruction algorithm (Griffin-lim, or even better in your case as you have a pretty decent estimate to begin with, using PGHI to correct phase imperfections) ?

**Strengths And Weaknesses:**

Overall, I feel that the topic of interpretability is of prime importance and the authors also nicely motivate their overall underlying motivations for this subject. The paper itself is nicely written and clearly exposes the different constituent aspects of the proposed approach. I think that the method is quite original and there are some interesting contributions to the interpretability field in general.

Regarding weaknesses, I think that my major concern is on the overall philosophy behind the approach.
- First, I feel that this approach could be quite easily compared to the "attribution" literature, which I did not see in the references. For instance, the following paper
Schulz, K., Sixt, L., Tombari, F., & Landgraf, T. (2020). Restricting the flow: Information bottlenecks for attribution. arXiv preprint arXiv:2001.00396.
https://arxiv.org/pdf/2001.00396.pdf
 Wouldn't it be simply possible to use such attribution methods as a spectral masking information in order to perform inverse STFT (akin to source separation) on the original spectrogram ? The strength of your approach is in uncovering the time activations matrices, but I feel that it is under-exploited in your results analysis.
- Second, I am wondering what would be the true use of the corresponding extracted sounds outside of the realm of simple classification tasks (eg. "car alarm" "dog barking"). I am unsure that this method could be insightful if it was applied to more complex tasks such as music genre classification ?

---

> ### Author Response · Authors · 2022-08-02
> **Response to Reviewer iYoX**
>
> Thank you for the positive review and interesting comments. We address your concerns in turn below.
>
> * **Attribution/Saliency map approaches**: Input attribution/saliency maps in their current form are more suitable for images. These maps are generally spatially smooth, which aids visual understandability, but are not effective masks to clearly emphasize time-frequency bins. Although, even for audio spectrogram like inputs, they can be useful in visually indicating the important regions, but are poor masks to filter such information for listenable output. We applied the mentioned approach (will add to references) to generate attribution maps for few samples on ESC50-Noise Experiment. Experiment details/results added on our companion website (<https://listen2interpret.000webhostapp.com/>).  While the saliency map visually indicates relevant regions, the time-domain signal still contains considerable noise and is not very useful. The smoothness of saliency maps can be partly attributed to upsampling of information extracted from lower resolution feature maps. Another limitation of applying these methods to 2D CNN's is their frequent use of log-mel spectrogram as input (current model uses 128 mel bands). The saliency map is then over the mel-spectrogram space. This adds to the loss of information and exacerbates issues in their use as filtering masks for spectrograms. Despite their usefulness, we believe these methods require non-trivial updates to be suitable for generating listenable interpretations.
>
> * **Applicability to other tasks**: Use of extracted parts of input audio (sound events, music, speech) as listenable interpretation could be very insightful for a number of audio detection, recognition, retrieval, transcription, or captioning tasks. DCASE challenges are a good indication of possible application areas for sound events. However, indeed, this may not be the most favorable form of interpreation for tasks such as music genre classification where interpretation in terms of high level aspects of input such as rhythm and tone is more meaningful. Also, music genre classification is not as well defined (in terms of categories) than sound events - annotators may clearly disagreee on the type of music genre for a given piece of music.
>
> * **Component matrix $\bf{W}$**: We believe the comment is incomplete. We will be happy to answer if you could please clarify. From what we understand: How good the dictionary $\bf{W}$ is, is related to how well the system can interpret. $\bf{W}$ is required to represent well various audio objects making up the input data.
>
> * **ESC50 performance + Applicability to other networks**: We will place the classifier's performance within the context of other popular works in literature. Please let us know if any additional details are needed. The main consideration in applying the current method to other networks is the design of $\Psi$ which processes hidden layers to output a dictionary-based representation. The current $\Psi$ was designed keeping in mind that it interprets CNNs operating on spectrogram-like representations (STFT/CQT/mel-spectras etc.). Intermediate layers of these CNN's have a notion of time and frequency. Given that we progressively downsample on frequency axis, the design could likely be extended to CNNs operating on raw-waveform which still carry a notion of time in their filter activations. Other network architectures (eg. Transformers, RNN) probably require significant redesigning of $\Psi$.
>
> * **Faithfulness evaluation**: Theoretically, range of FF is $(-\infty, \infty)$ as the logit value can be anywhere on the real line. The average logit value for the predicted class (over 5 folds) is 5.77. For better context, we also computed median probability drop. The paper currently contains $FF_{median}$ over ESC-50 (fold 1) for $\tau$ from $0.9$ to $0.3$. We'll update these results by reporting average of $\text{FF}_\text{median}$ over 5 folds for logit drop and probability drop from $\tau$ $0.9$ to $0.1$. We also updated the 'Random Baseline' to randomly select components to remove from among all the components. The results are given in a table below. Last row in the table is for 'Random Baseline':
> | Threshold $\tau$  | $\text{FF}_\text{median, logit}$  | $\text{FF}_\text{median, prob}$  |
> | ------------- |:-----:| -----:|
> | $\tau=0.9$    | 0.336 | 0.002 |
> | $\tau=0.7$    | 0.526 | 0.004 |
> | $\tau=0.5$    | 0.854 | 0.012 |
> | $\tau=0.3$    | 1.374 | 0.040 |
> | $\tau=0.1$    | 2.126 | 0.113 |
> | -------- |-----|-----|
> | $\tau=0.1$    | 0.015 | $<10^{-4}$ |
>
> * **Phase information for time-domain inversion**: Yes, we use phase of original input spectrogram for time-domain inversion since this has been a very common practice in the NMF and source separation literature. We will make this more explicit in Sec 3.2. As pointed out, one could employ a phase estimation algorithm to possibly improve over our time-domain conversion strategy. We will add this as a limitation.

---

### Official Review · Reviewer_1KUh · 2022-07-13

**Rating:** 7
**Confidence:** 4
**Soundness:** 3 good
**Presentation:** 3 good
**Contribution:** 3 good

**Summary:**

The paper presents a novel approach for interpreting trained neural network models using audio as input. The approach is based on the NMF technique, and is using two neural networks: one to estimate the time activation of the NMF from the hidden representations of the trained model and one to approximate the output of the trained model using the time activations. The approach is evaluated on the SED task on two corpora. Two novel quantitative metrics are proposed, fidelity and faithfulness, and the proposed approach is compared to other audio interpretation baselines. The results show that the proposed approach is on par with the state-of-the-art while being able to provide listenable interpretations.

**Questions:**

- About the writing:
    - I think the pre-learn NMF dictionary should be described more clearly in the main paper instead of being hidden in the Supplementary material, at least briefly.
    - I don't really like the use of symbols as names for the two models in the Interpreter (\Psi and \Theta), it would helpful if they have a proper name, something like the "Generator" for Psi and the "Approximator" for Theta.
- The proposed approach is a bit similar to the idea of "relevance map", i.e. finding what part of the input is the most useful for the classificaition, for instance in [1], the author should discuss that kind of approach.

[1] Muckenhirn, Hannah, Vinayak Abrol, Mathew Magimai-Doss, and Sébastien Marcel. "Understanding and Visualizing Raw Waveform-Based CNNs." In Interspeech, pp. 2345-2349. 2019.

**Limitations:**

The limitations are only addressed in the Supplementary material and not in the main paper. It's also quite short, the authors should discuss the performance of the interpreter on out-of-domain data for instance.

**Strengths And Weaknesses:**

Strengths:
- The paper is generally well written and easy to follow.
- The approach is clearly novel, especially for audio interpretability.
- The experiments are well designed and thorough, clearly showing the capabilities of the approach. The provided audio examples are interesting and illustrate the approach well. The subjective evaluation is a very welcome addition.
- The proposed approach is a good step towards more interpretability in neural networks, specially on the audio modality.

Weaknesses:
- The significance of the approach is limited, because:
    - It is not clear if it would be applicable to other types of audio data and tasks, such as speech recognition, or to other modalities like images.
    - I would generally hope that interpretability papers also bring new insights on what the neural networks have learned, but it's not the case in this paper, it's just showing that the model is doing what is expected, i.e. discard obvious irrelevant information. It would be more interesting to apply the method on other tasks where it's not obvious which part of the input is relevant to the task, for instance Speech Emotion Recognition.
- The writing could be improved (see below).
- Missing references in the related work.

Overall, I recommend this paper for acceptance as the approach is novel and could be of interest of the NeurIPS community.

---

> ### Author Response · Authors · 2022-08-02
> **Response to Reviewer 1KUh**
>
> Thank you for the positive comments. We respond to your questions below
>
> * **Method's applicability**: Use of extracted parts of input audio (sound events, music, speech) as listenable interpretation could
> be very insightful for a number of audio detection, recognition, retrieval,
> transcription, or captioning tasks. DCASE challenges are a good indication of possible application areas for sound events. However, this may not be the most favorable form of interpretation for tasks such as music genre classification where the mapping between categories and underlying audio object/events is not as clear. Although the decomposition-based design methodology could potentially be extended to other modalities such as images or text, the method in its current form is specifically designed for the audio modality.
>
> * **Interpretability for new insights into NN**: We agree with the reviewer and it would be interesting to try it on new tasks, including the one suggested. Our qualitative mis-classification experiment on ESC50 (discussion in supplement Sec A.1.5.2, examples on companion website) are aligned with the suggestion where our interpretations provide useful insights when confusion arises between acoustically similar classes.
>
> * **Suggested reference**: While we do share some similarities with attribution/saliency map approaches in terms of final interpretation, by identifying parts of input relevant for decision, we carry significant methodological differences. Our formulation is comparatively much closer to "concept-based approaches". We will include the suggested reference in Sec. 2. It applies guided backpropagation on CNNs which take raw time-domain signals as input to extract a "relevance signal". This 1D relevance signal is analyzed in spectral domain to bring out meaningful insights. The same analysis is not transferable for 2D spectrogram-like inputs as in our current case.
>
> * **Writing changes suggested**: Thanks for the remarks. We will take them into account in our final version.
>
> * **Out-of-domain data**: We will move the limitations to main paper. The topics of domain adaptation and covariate shift for interpretations are interesting for further research. We will also include this discussion. Specifically, in regard to interpreter's performance on out-of-domain data, we expect that they will be closely linked to those of the classifier on such data.

---

### Meta-Review · Area_Chair_g2Rh · 2022-08-24

**Recommendation:** Accept
**Confidence:** Certain

**Metareview:**

The paper presents a novel approach for interpreting a neural network's decision on audio input. The motivation of the research is clear, and all the reviewers have agreed on the importance of the addressed task and the originality of the proposed solution. Also, the authors have adequately responded to the reviewers' concerns during the rebuttal, including some additional experiments such as a comparison with attribution interpretation or a modified baseline. Therefore, I gladly recommend this paper be accepted in NeurIPS 2022.

## Strengths
- The paper addresses one of the underexplored yet important tasks of interpretability of neural networks for the audio domain. As visualization is helpful for interpreting neural networks in the vision domain, sonification is a reasonable and intuitive way to analyze how the neural network works on audio. This paper presents a promising solution for this purpose.
- The architecture of the proposed model is original and shows high novelty. The experiment for the evaluation was well-designed, showing meaningful results.

## Weaknesses
- Many reviewers suggested that it would be interesting to see how the proposed model works on more complex data, such as music audio. The authors have explained why they focused on sound event detection as an initial goal. The reviewer's suggestion would remain a challenging but valuable goal for future research.


**Award:**

No

---

### Decision · Program_Chairs · 2022-09-14

Accept